# Uncertainty-aware Active Learning for Optimal Bayesian Classifier

**Guang Zhao[1], Edward R. Dougherty[1], Byung-Jun Yoon[1,3], Francis J. Alexander[3], & Xiaoning Qian[1,2]**
guangzhao@tamu.edu, falexander@bnl.gov,
{edward,bjyoon,xqian}@ece.tamu.edu

[1]**Department of Electrical & Computer Engineering,**
[2]**Department of Computer Science & Engineering,**
Texas A&M University
College Station, TX 77843, USA

[3]**Computational Science Initiative,**
Brookhaven National Laborator
Upton, NY 11973, USA

## ABSTRACT

For pool-based active learning, in each iteration a candidate training sample is chosen for labeling by optimizing an acquisition function. In Bayesian classification, expected Loss Reduction (ELR) methods maximize the expected reduction in the classification error given a new labeled candidate based on a one-step-look-ahead strategy. ELR is the optimal strategy with a single query; however, since such myopic strategies cannot identify the long-term effect of a query on the classification error, ELR may get stuck before reaching the optimal classifier. In this paper, inspired by the mean objective cost of uncertainty (MOCU), a metric quantifying the uncertainty directly affecting the classification error, we propose an acquisition function based on a weighted form of MOCU. Similar to ELR, the proposed method focuses on the reduction of the uncertainty that pertains to the classification error. But unlike any other existing scheme, it provides the critical advantage that the resulting Bayesian active learning algorithm guarantees convergence to the optimal classifier of the true model. We demonstrate its performance with both synthetic and real-world datasets.

## 1 INTRODUCTION

In supervised learning, labeling data is often expensive and highly time consuming. Active learning is one field of research that aims to address this problem and has been demonstrated for sample-efficient learning with less required labeled data (Gal et al., 2017; Tran et al., 2019; Sinha et al., 2019). In this paper, we focus on pool-based Bayesian active learning for classification with 0-1 loss function. Bayesian active learning starts from the prior knowledge of uncertain models. By optimizing an acquisition function, it chooses the next candidate training sample to query for labeling, and then based on the acquired data, updates the belief of uncertain models through Bayes' rule to approach the optimal classifier of the true model, which minimizes the classification error.

In active learning, maximizing the performance of the model trained on queried candidates is the ultimate objective. However, most of the existing methods do not directly target the learning objective. For example, Maximum Entropy Sampling (MES) or Uncertainty Sampling, simply queries the candidate with the maximum predictive entropy (Lewis & Gale, 1994; Sebastiani & Wynn, 2000; Mussmann & Liang, 2018); but the method fails to differentiate between the model uncertainty and the observation uncertainty. Bayesian Active Learning by Disagreement (BALD) seeks the data point that maximizes the mutual information between the observation and the model parameters (Houlsby et al., 2011; Kirsch et al., 2019). Besides BALD, there are also other methods reducing the model uncertainty in different forms (Golovin et al., 2010; Cuong et al., 2013). However, not all the model uncertainty will affect the performance of the learning task of interest. Without identifying whether the uncertainty is related to the classification error or not, these methods can be inefficient in the sense that it may query candidates that do not directly help improve prediction performance.

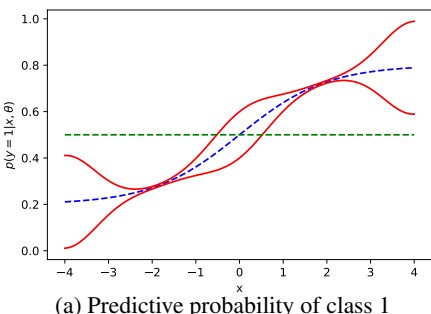
(a) Predictive probability of class 1

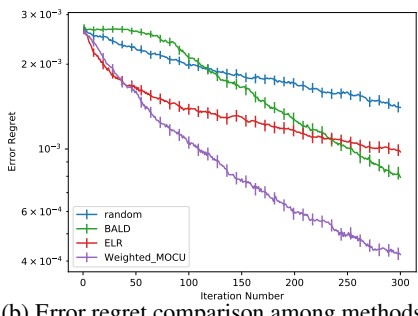
(b) Error regret comparison among methods

Figure 1: (a) Predictive probability of class 1 under uncertainty: the red lines indicate the upper and lower bounds of the predictive probability; the blue dash line is the mean of the predictive probability; the green dash line indicates that the probability is equal to 0.5. (b) Active learning performance comparison.

In this paper we focus on the active learning methods directly maximizing the learning model performance. There exist such active learning methods by Expected Loss Reduction (ELR) that aim to maximize the expected reduction in loss based on a one-step-look-ahead manner (Roy & McCallum, 2001; Zhu et al., 2003; Kapoor et al., 2007). The ELR methods can focus on only the uncertainty related to the loss function to achieve sample-efficient learning. In fact, ELR is the optimal strategy for active learning with a single query (Roy & McCallum, 2001). However, a critical shortcoming of previous ELR schemes is that none of them provide any theoretical guarantee regarding their long-term performance. In fact, since these methods are myopic and cannot identify the long-term effect of a query on the loss functions, without special design on the loss function, they may get stuck before reaching the optimal classifier. To the best of our knowledge, there is currently no method that directly maximizes the model performance while simultaneously guaranteeing the convergence to the optimal classifier.

Fig. 1a provides an example of binary classification with one feature where both BALD and ELR methods fail. In the figure, the red lines indicate the upper and lower bounds of the prediction probability of class 1, illustrating the model with higher probability uncertainty on the sides ($x \to \pm 4$) than that in the middle ($x = 0$). Querying candidates on the sides will provide more information of the model parameters, and therefore is preferred in BALD. However, since the possible probabilities on the sides are always larger than or less than 0.5, querying candidates on the sides will not help reduce the classification error. On the other hand, ELR queries candidates that help reduce the classification error the most, so it prefers data in the middle whose optimal labels are uncertain given the prior knowledge. The performance shown in Fig. 1b agrees with our analysis. Fig. 1b shows the performance averaged over 1000 runs, with more details and discussions of the example included in Appendix C. BALD performs inefficiently at the beginning by querying points on both sides. On the other hand, the ELR method performs the best at the beginning, but becomes inefficient after some iterations ($\sim$100), indicating some of its runs get stuck before reaching the optimal classifier. In this paper, we consider the algorithm to "get stuck" when the acquisition function value is 0 for all the candidates in the pool and the algorithm degenerates to uniform random sampling.

In this paper, we analyze the reason why ELR methods may get stuck before reaching the optimal classifier, and propose a new strategy to solve this problem. Our contributions are in four parts: 1. We show that ELR methods may get stuck, preventing active learning from reaching the optimal classifier efficiently. 2. We propose a novel weighted-MOCU active learning method that can focus only on the uncertainty related to the loss for efficient active learning and is guaranteed to converge to the optimal classifier of the true model. 3. We provide the convergence proof of the weighted-MOCU method. 4. We demonstrate the sample-efficiency of our weighted-MOCU method with both synthetic and real-world datasets.

## 2 BACKGROUND

**Optimal Bayesian classifier.** Consider a classification problem with candidates $x \in \mathcal{X}$ and class labels $y \in \mathcal{Y} = \{0, 1, \ldots, M-1\}$. The predictive probability $p(y|x, \theta)$ is modeled with parameters

$\theta$. Assume $\theta$ is uncertain with a distribution $\pi(\theta)$ within the uncertainty class $\Theta$. The classification problem is to find a classifier $\psi : \mathcal{X} \rightarrow \mathcal{Y}$, which assigns a predicted class label to a given candidate.

The expected 0-1 loss of the classifier $\psi$ for a candidate $x$, dependent on $\theta$, is defined as $C_\theta(\psi, x)$, which can be derived to be the classification error: $C_\theta(\psi, x) = 1 - p(y = \psi(x)|x, \theta)$. The *optimal classifier* with $\theta$, $\psi_\theta$ is defined as the classifier minimizing the classification error: $\psi_\theta(x) = \arg\max_y p(y|x, \theta)$. So we have: $C_\theta(\psi_\theta, x) = \min_\psi C_\theta(\psi, x) = \min_y\{1 - p(y|x, \theta)\}$. When there is model uncertainty with $\pi(\theta)$, an Optimal Bayesian Classifier (OBC) $\psi_{\pi(\theta)}$ is the classifier that has the minimum expected loss over $\pi(\theta)$ (Dalton & Dougherty, 2013):

$$\mathbb{E}_{\pi(\theta)}[C_\theta(\psi_{\pi(\theta)}, x)] = \min_\psi \mathbb{E}_{\pi(\theta)}[C_\theta(\psi, x)] = \min_y\{1 - p(y|x)\} \tag{1}$$

where $p(y|x) = \mathbb{E}_{\pi(\theta)}[p(y|x, \theta)]$ is the predictive distribution. It's easily to see $\psi_{\pi(\theta)}(x) = \arg\max_y p(y|x)$.

**Active learning.** Active learning collects the training dataset $D$ in a sequential way. For pool-based active learning, in each iteration, we choose a candidate $x$ from the set of potential training samples $\mathcal{X}$ to query for the class label by optimizing an acquisition function $U(x)$. Then, in the Bayesian setting, by including the observed data pair $(x, y)$ to $D$, we update the posterior distribution based on Bayes' rule. In each iteration, the acquisition function depends on the posterior distribution of model parameters $\pi(\theta|D)$. In the following discussion, to simplify notations, we omit $D$ from the notations and use $\pi(\theta)$ and $p(y|x)$ to respectively denote the posterior and predictive distributions conditioned on $D$. When a new observed data point is included, the distributions are updated by Bayes' rule and the total probability rule as: $\pi(\theta|x, y) = \frac{\pi(\theta)p(y|x, \theta)}{p(y|x)}$ and $p(y'|x', x, y) = \mathbb{E}_{\pi(\theta|x, y)}[p(y'|x', \theta)]$.

The acquisition function of ELR methods in the Bayesian setting can be defined by the expected OBC prediction error reduction after observing the new pair $(x, y)$ (Roy & McCallum, 2001):

$$U^{\text{ELR}}(x) = \mathbb{E}_{p(x')}\{\mathbb{E}_{\pi(\theta)}[C_\theta(\psi_{\pi(\theta)}, x')] - \mathbb{E}_{p(y|x)}[\mathbb{E}_{\pi(\theta|x, y)}[C_\theta(\psi_{\pi(\theta|x, y)}, x')]]\}, \tag{2}$$

where $p(x')$ is the distribution over $\mathcal{X}$, independent of $\theta$ and $D$. ELR methods assume that we use OBC as the classifier, and in each iteration we should choose the query that maximize the decrease in OBC prediction error. The first term in (2) is the OBC prediction error of $\psi_{\pi(\theta)}$, and the second term is the expected prediction error of $\psi_{\pi(\theta|x, y)}$, the one-step-look-ahead OBC, with respect to $p(y|x)$. In the following section, we analyze why this acquisition function is sample-efficient as it directly targets at classification error reduction while ignoring irrelevant uncertainty with respect to the learning task; but it may get stuck before converging to the true optimal classifier (optimal classifier of the true model).

## 3 MOCU-BASED ACTIVE LEARNING

### 3.1 MEAN OBJECTIVE COST OF UNCERTAINTY

To analyze ELR methods, we borrow the idea of the Mean Objective Cost of Uncertainty (MOCU) for active learning with respect to the corresponding posterior $\pi(\theta)$. MOCU is a general objective-oriented uncertainty quantification framework (Yoon et al., 2013). For active learning, MOCU can be defined as the expected loss difference between the OBC and the optimal classifier:

$$\mathcal{M}(\pi(\theta)) = \mathbb{E}_{p(x')}[\mathbb{E}_{\pi(\theta)}[C_\theta(\psi_{\pi(\theta)}, x') - C_\theta(\psi_\theta, x')]] \tag{3}$$

$$= \mathbb{E}_{p(x')}[\min_{y'}\{1 - p(y'|x')\}] - \mathbb{E}_{\pi(\theta)}[\min_{y'}\{1 - p(y'|x', \theta)\}]]. \tag{4}$$

The second line is derived by the definition of $\psi_\theta$ and (1). The first term in (3) is the OBC error as the loss. In the second term, $\psi_\theta$ is the optimal classifier with a specific $\theta$. For the terms inside the expectation operator, we have $C_\theta(\psi_{\pi(\theta)}, x') - C_\theta(\psi_\theta, x') \geq 0$. Therefore, the second term in (3) is a lower bound of the OBC prediction error. MOCU captures the difference between the OBC error and its lower bound. When MOCU is 0, the OBC converges to the true optimal classifier and we cannot reduce the OBC prediction error further. In that case, we say that OBC has reached the true optimal classifier.

As in ELR methods, we can define an acquisition function by the reduction of MOCU in a one-step-look-ahead manner:

$$U^{\text{MOCU}}(x; \pi(\theta)) = \mathcal{M}(\pi(\theta)) - \mathbb{E}_{p(y|x)}[\mathcal{M}(\pi(\theta|x, y))]. \tag{5}$$

We can show that the second term in (3), the lower bound of the OBC error, is cancelled in (5). The acquisition function (5) hence captures the expected reduction of the OBC error given new data and is equivalent to the ELR acquisition function (2). Expanding the second term in (5), we have:

$$\mathbb{E}_{p(y|x)}[\mathcal{M}(\pi(\theta|x, y))] = \mathbb{E}_{p(x')}\{\mathbb{E}_{p(y|x)}[\mathbb{E}_{\pi(\theta|x,y)}[C_\theta(\psi_{\pi(\theta|x,y)}, x') - C_\theta(\psi_\theta, x')]]\}. \tag{6}$$

Since $\sum_y p(y|x)\pi(\theta|x, y) = \pi(\theta)$, as $x$ is assumed to be independent of $\theta$ so that we have $\pi(\theta|x) = \pi(\theta)$, we can rewrite the first term in (5) as:

$$\mathcal{M}(\pi(\theta)) = \mathbb{E}_{p(x')}\{\mathbb{E}_{p(y|x)}[\mathbb{E}_{\pi(\theta|x,y)}[C_\theta(\psi_{\pi(\theta)}, x') - C_\theta(\psi_\theta, x')]]\}. \tag{7}$$

Combining (6) and (7) and canceling the $C_\theta(\psi_\theta, x')$ terms (the lower bound of the OBC error), (6) can be derived as:

$$U^{\text{MOCU}}(x; \pi(\theta)) = \mathbb{E}_{p(x')}\{\mathbb{E}_{p(y|x)}[\mathbb{E}_{\pi(\theta|x,y)}[C_\theta(\psi_{\pi(\theta)}, x') - C_\theta(\psi_{\pi(\theta|x,y)}, x')]]\}, \tag{8}$$

which is just the ELR acquisition function in (2). Therefore, we can conclude that MOCU-based methods are equivalent to ELR methods.

Another property we can observe from (8) is that $U^{\text{MOCU}}(x; \pi(\theta)) \geq 0$. By definition, $\psi_{\pi(\theta|x,y)}$ is the OBC with the minimum expected classification error over $\pi(\theta|x, y)$. Therefore, $\mathbb{E}_{\pi(\theta|x,y)}[C_\theta(\psi_{\pi(\theta|x,y)}, x')] \leq \mathbb{E}_{\pi(\theta|x,y)}[C_\theta(\psi_{\pi(\theta)}, x')]$ and we have $U^{\text{MOCU}}(x; \pi(\theta)) \geq 0$, indicating collecting new data will reduce MOCU.

## 3.2 ANALYSIS OF ELR METHODS

In the following analysis, we assume that $\Theta$ contains the true model $\theta_r$ and $\pi(\theta_r) > 0$. We first analyze ELR methods by the MOCU reduction to show that ELR and MOCU-based active learning ignores the uncertainty irrelevant to the OBC prediction. By that, we indicate that not all the model uncertainties directly affect the OBC prediction. Denote the contribution to the MOCU at point $x$ as $K(x, \pi(\theta)) = \mathbb{E}_{\pi(\theta)}[C_\theta(\psi_{\pi(\theta)}, x) - C_\theta(\psi_\theta, x)]$, so that $\mathcal{M}(\pi(\theta)) = \mathbb{E}_{p(x)}[K(x, \pi(\theta))]$. If $K(x, \pi(\theta)) = 0$, then we have $\forall \theta \in \text{supp}(\pi)$, $\psi_\theta(x) = \psi_{\pi(\theta)}(x)$, i.e. $\arg\max_y p(y|x, \theta) = \arg\max_y p(y|x)$. This means that for all the possible models, the optimal predictions are the same, and the OBC prediction on $x$ will not be affected by the remaining uncertainty of $p(y|x, \theta)$, if any. In fact, $K(x, \pi(\theta)) = 0$ does not necessarily mean that there is no uncertainty associated with $p(y|x, \theta)$, for which it requires that the value of $p(y|x, \theta)$ is the same $\forall \theta \in \text{supp}(\pi)$, apparently a stronger statement than $K(x, \pi(\theta))$ being 0. Therefore, not all the uncertainties of $p(y|x, \theta)$ are captured in MOCU when $K(x, \pi(\theta)) = 0$. We consider the uncertainty in $p(y|x, \theta)$ to be "objective-irrelevant" to the OBC prediction if $K(x, \pi(\theta)) = 0$. In the active learning procedure, when a new observation is obtained, it reduces the uncertainty of the parameter $\theta$; and as a result, it reduces the uncertainty of $p(y|x, \theta)$ for each $x \in \mathcal{X}$. If an observation only reduces objective-irrelevant uncertainty, the value of MOCU will not change. For example, in Fig. 1a, the uncertainty of $p(y|x, \theta)$ in the region close to $x = \pm 4$ is objective-irrelevant. Evaluating points at $x \to \pm 4$ will only reduce irrelevant uncertainty and it will not be considered in either MOCU- or ELR-based methods. That explains why in the first several active learning iterations, the ELR or MOCU-based active learning can be more efficient than the methods guided by total uncertainty reduction, such as BALD.

Now we explain why ELR methods may get stuck before the OBC converges to the true optimal classifier. When we have $\forall x \in \mathcal{X}, U^{\text{MOCU}}(x; \pi(\theta)) = 0$ and assume the tie is broken randomly, the acquisition function will suggest any random candidate in the pool. When that happens, we say that ELR methods get stuck if the OBC has not reached the true optimal classifier; i.e., $\mathcal{M}(\pi(\theta))$ is still larger than 0.

Since $p(y'|x') = \mathbb{E}_{\pi(\theta)}[p(y'|x', \theta)]$ is a linear function of $\pi(\theta)$, the term $\min_{y'}\{1 - p(y'|x')\}$ in (4) is the minimum among $M$ linear functions, and thus a concave piece-wise linear function. Within each linear piece, $\psi_{\pi(\theta)}(x') = \arg\max_{y'} p(y'|x')$ are the same for different $\pi(\theta)$. The second term $\mathbb{E}_{\pi(\theta)}[\min_{y'}\{1 - p(y'|x', \theta)\}]$ in (4) is a linear function of $\pi(\theta)$. Subtracting it from the first term and averaging the resulting difference over $p(x')$ maintain the concavity and the piece-wise linearity.

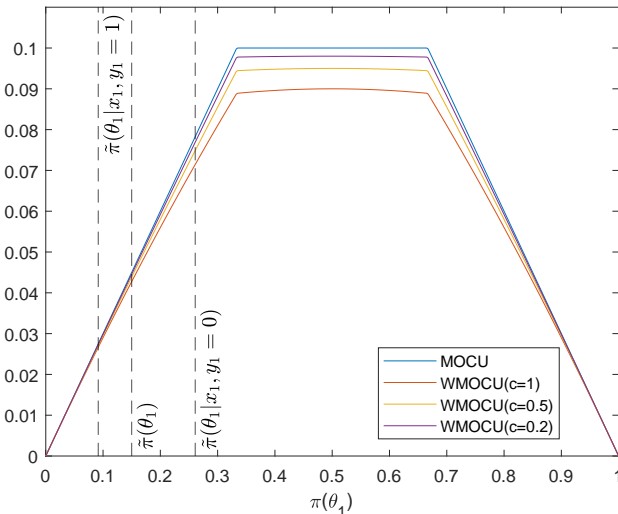

Figure 2: MOCU and weighted-MOCU functions of a binary classification problem with $\Theta = \{\theta_1, \theta_2\}$.

Therefore, MOCU defined in (4) is a concave piece-wise linear function of $\pi(\theta)$. Moreover, within a linear piece of MOCU, $\psi_{\pi(\theta)}(x') = \arg\max_{y'} p(y'|x')$, i.e. the OBC prediction, is the same for each of $x' \in \mathcal{X}$. To gain some intuition, we study a binary classification problem with the uncertainty class of two possible models $\Theta = \{\theta_1, \theta_2\}$ and a candidate pool of two training samples to query: $\mathcal{X} = \{x_1, x_2\}$. Further details of the model setup can be found in Appendix D. Since $\pi(\theta_1) = 1 - \pi(\theta_2)$, we can express the MOCU function as a univariate function of $\pi(\theta_1)$ as shown in Fig. 2. It is clear that the MOCU function is a concave piece-wise linear function.

Since $\pi(\theta) = \mathbb{E}_{p(y|x)}[\pi(\theta|x,y)]$, from (5), the acquisition function is defined as $\mathcal{M}[\mathbb{E}_{p(y|x)}[\pi(\theta|x,y)]] - \mathbb{E}_{p(y|x)}[\mathcal{M}(\pi(\theta|x,y))]$. Based on the concavity of MOCU $\mathcal{M}(\cdot)$ and Jensen's inequality, we have the acquisition function $U^{\text{MOCU}}(x; \pi(\theta)) \geq 0$. The equality holds if for all $y \in \mathcal{Y}$, $\pi(\theta|x,y)$ falls into the same linear piece of MOCU. In this case, the single query $(x,y)$ cannot provide enough evidence to shift the OBC prediction, so that $\arg\max_{y'} p(y'|x',x,y) = \arg\max_{y'} p(y'|x')$ for each of $x' \in \mathcal{X}$, even though $p(y'|x',x,y) \neq p(y'|x')$. In Fig. 2, we have shown in the binary classification problem there exists such a case that $\tilde{\pi}(\theta_1|x_1, y_1 = 0)$ and $\tilde{\pi}(\theta_1|x_1, y_1 = 1)$ are within the interval of the same linear piece of the corresponding MOCU function. When MOCU is larger than 0, if all candidates cannot provide enough evidence to change the OBC predictions, ELR and MOCU-based methods will get stuck before converging to the true optimal classifier. This is due to the myopic nature of the acquisition function ignoring the long-term effect of querying each candidate.

In summary, ELR methods are efficient by ignoring objective-irrelevant uncertainty. But if sampling one data point can only provide little information and cannot help improve OBC prediction in the current iteration, they will ignore its long-term effect on prediction performance. As a result, they may get stuck before converging to the optimal classifier. To keep the efficient sampling property by ignoring objective-irrelevant uncertainty but avoid getting stuck, we propose a one-step-look-ahead acquisition function based on a weighted version of MOCU, which can capture the change of the predictive probability from one-step query candidates. If that change can potentially shift the OBC predictions in the long run, our acquisition function will have a positive value, and thereby avoid the issues that ELR methods suffer from.

### 3.3 WEIGHTED MOCU-BASED ACTIVE LEARNING

In this section, we propose a modified MOCU-based acquisition function that has the theoretical guarantee to converge to the optimal classifier. Specifically, we propose a modified MOCU function that multiplies a weight with each loss difference between the OBC $\psi_{\pi(\theta)}$ and the optimal classifier

$\psi_\theta$ in the original MOCU definition:

$$\mathcal{M}^w(\pi(\theta)) = \mathbb{E}_{p(x')}\{\mathbb{E}_{\pi(\theta)}\{w(\pi(\theta), x', \theta)[C_\theta(\psi_{\pi(\theta)}, x') - C_\theta(\psi_\theta, x')]\}\}, \qquad (9)$$

where $w(\pi(\theta), x', \theta) > 0$ is the weighting function. The corresponding acquisition function is:

$$U^w(x; \pi(\theta)) = \mathcal{M}^w(\pi(\theta)) - \mathbb{E}_{p(y|x)}[\mathcal{M}^w(\pi(\theta|x, y))]. \qquad (10)$$

In (9), as more data are collected and the model parameter distribution $\pi(\theta)$ changes, $w(\pi(\theta), x', \theta)$ will change accordingly. The change of $w(\pi(\theta), x', \theta)$ cannot affect the value of the weighted MOCU if $C_\theta(\psi_{\pi(\theta)}, x') - C_\theta(\psi_\theta, x') = 0$, $\forall \theta \in \text{supp}(\pi(\theta))$, indicating the uncertainty at $x'$ is objective-irrelevant. This makes sure that the acquisition function based on the weighted MOCU will inherit the property of MOCU-based active learning to directly target at classification error reduction while ignoring irrelevant uncertainty. On the other hand, by introducing the predictive probability into the weighting functions, the probability change from one-step samples can be captured by the weighted-MOCU based acquisition function such that it can have theoretical convergence guaranteed to the optimal classifier as shown below.

We would like to emphasize that there are also active learning algorithms, such as the ones based on the cyclic sampling and $\epsilon$-greedy policies (Hoang et al., 2014), that can almost surely converge to the true model, and as a result, the OBC converges to the true optimal classifier. However, these policies focus on the *total* uncertainty reduction to derive the *full* knowledge of the true model, which is unnecessary and therefore inefficient, since we only need the knowledge of the true optimal classifier if the classification performance is the primary concern. Unlike such policies, our weighted-MOCU based policy directly reduces the *objective* uncertainty affecting classification, and as a result, it is much more efficient by focusing only on those queries that are helpful for improving the prediction. As a result, our proposed algorithm guarantees efficiency both in the short term as well as in the longer term.

In the following, we design a weighting function to make $\mathcal{M}^w(\pi(\theta)) = 0$ if and only if $\forall x \in \mathcal{X}, U^w(x; \pi(\theta)) = 0$ and show that active learning based on this weighted MOCU converges to the optimal classifier. Specifically, we propose the following weighting function:

$$w(\pi(\theta), x', \theta) = 1 - c \cdot K(x', \pi(\theta)), \text{ with} \qquad (11)$$

$$K(x', \pi(\theta)) = \mathbb{E}_{\pi(\theta)}[C_\theta(\psi_{\pi(\theta)}, x') - C_\theta(\psi_\theta, x')] \qquad (12)$$

$$= \min_{y'} \mathbb{E}_{\pi(\theta)}[1 - p(y'|x', \theta)] - \mathbb{E}_{\pi(\theta)}[\min_{y'}(1 - p(y'|x', \theta)] \qquad (13)$$

$$= \mathbb{E}_{\pi(\theta)}[\max_{y'} p(y'|x', \theta)] - \max_{y'} p(y'|x'), \qquad (14)$$

where $0 < c \leq 1$ is a parameter controlling the approximation of the weighted MOCU to the original MOCU, with smaller $c$ giving a better approximation. The choice of $c$ depends on the specific classification problem and the total query budget. Methods using a smaller $c$ approximate the ELR methods better, hence they will perform well in the first several iterations but may converge slowly in the long run. On the other hand, when $c$ is closer to 1, the acquisition function weighs more heavily on long-term benefits. It is clear that $K(x', \pi(\theta)) \geq 0$ by (13). For binary classification, $\max_{y'} p(y'|x') \geq 0.5$. As $\mathbb{E}_{\pi(\theta)}[\max_{y'} p(y'|x', \theta)] \leq 1$, from (14), we have $K(x', \pi(\theta)) \leq 0.5$, demonstrating that the weighting function in (11) satisfies the requirement $w(\pi(\theta), x', \theta) \geq 0.5 > 0$.

Note that this simple weighting function does not change with respect to the model parameter values. Substituting it into the weighted MOCU expression, we have:

$$\mathcal{M}^w(\pi(\theta)) = \mathbb{E}_{p(x')}\{(1 - cK(x', \pi(\theta))) \cdot K(x', \pi(\theta))\}, \qquad (15)$$

which is a strictly concave function of $K$. We also illustrate the weighted-MOCU function in Fig. 2 for the same example in Section 3.2. As shown in the figure, a smaller $c$ provides better approximation to the MOCU function, and all the weighted MOCU functions are strictly concave functions of $\pi(\theta_1)$ instead of being piece-wise linear, which guarantees that the acquisition function $U^w(x_1; \tilde{\pi}(\theta))$ is positive. In general, weighted MOCU is strictly concave along most of the directions and only changes linearly along the directions that $K(x, \pi(\theta))$ is constant for $x \in \mathcal{X}$, which correspond to the queries that only reduce irrelevant uncertainties. Such a property can guarantee the convergence to the true optimal classifier.

Before presenting the theoretical convergence guarantee of the weighted-MOCU based active learning, we summarize the computation of our weighted-MOCU based acquisition function in Algorithm 1, which can replace ELR and MOCU-based acquisition functions in Bayesian active learning algorithms with the pseudo-code given in Appendix B. We estimate the computational complexity of Algorithm 1 for the discrete feature and parameter spaces. Assume that the size of the discrete feature space is $N_x = |\mathcal{X}|$ and the size of the uncertainty set of classifiers is $N_\theta = |\Theta|$. We study the complexity of calculating the weighted MOCU. In the WMOCU function, the OBC error evaluation in line 19 is called for $O(N_x N_\theta)$ times. In ACQUISITIONFUN, WMOCU is called for constant times. Hence, the total complexity of calculating the acquisition function in weighted-MOCU based active learning is $O(N_x N_\theta)$. Compared with the ELR method, there is $O(N_x)$ additional computation associated with computing the weight $(1 - cK)$ in line 26. Hence, the incurred computational complexity is of the same order as the original ELR and MOCU-based methods.

---

**Algorithm 1** Calculation for Weighted-MOCU based Acquisition Function

---

1: **function** ACQUISITIONFUN($x, \pi_{\theta|D}, c$)
2:      $wmocu\_current =$ WMOCU($\pi_{\theta|D}$)
3:      $wmocu\_next = 0$
4:      **for** $y$ in $\{0, 1\}$ **do**
5:          **for** $\theta$ in $\Theta$ **do**
6:              Generate array $p(\theta, y|D, x) = \pi_{\theta|D} \cdot p(y|x, \theta)$
7:          **end for**
8:          $p(y|D, x) = \sum_\theta p(\theta, y|D, x)$
9:          $\pi_{\theta|D,x,y} = p(\theta, y|D, x)/p(y|D, x)$
10:          $wmocu\_next = wmocu\_next + p(y|D, x) \cdot$ WMOCU($\pi_{\theta|D,x,y}, c$)
11:      **end for**
12:      **return** $wmocu\_current - wmocu\_next$
13: **end function**

14: **function** WMOCU($\pi_{\theta|D}, c$)
15:      $wmocu = 0$
16:      **for** $x'$ in $\mathcal{X}$ **do**
17:          $bayesian\_error = 0$
18:          **for** $\theta$ in $\Theta$ **do**
19:              $bayesian\_error = bayesian\_error + \pi_{\theta|D} \cdot (1 - \max_{y'} p(y'|x', \theta))$
20:          **end for**
21:          **for** $y'$ in $\{0, 1\}$ **do**
22:              $p(y'|D, x') = \sum_\theta \pi_{\theta|D} \cdot p(y'|x', \theta)$
23:          **end for**
24:          $obc\_error = 1 - \max_{y'} p(y'|D, x')$
25:          $K = obc\_error - bayesian\_error$
26:          $wmocu = wmocu + p(x') \cdot [(1 - cK)K]$
27:      **end for**
28:      **return** $wmocu$
29: **end function**

---

**Theoretical convergence guarantee.** Now we show that if active learning for a binary classification problem is guided by the acquisition function defined by (10) and (11), MOCU will converge to 0 almost surely and hence the procedure will converge to learning the optimal classifier of the true model. We assume that both $\mathcal{X}$ and $\Theta$ are discrete with finite elements; the true model parameter $\theta_r \in \Theta$ and the prior distribution $\pi^0(\theta)$ over $\Theta$ satisfies $\pi^0(\theta_r) > 0$. We denote the posterior by $\pi^n(\theta)$ and predictive probability $p^n(y|x)$ in the $n$-th weighted MOCU based active learning iteration, respectively. In the following, we give important lemmas first. All the proofs of the presented lemmas can be found in Appendix A.

**Lemma 1** *Given $\pi(\theta)$, $\mathcal{M}(\pi(\theta)) = 0$ if and only if $\mathcal{M}^w(\pi(\theta)) = 0$.*

Lemma 1 indicates that if $\mathcal{M}^w(\pi(\theta)) = 0$, the OBC $\psi_{\pi(\theta)}$ converges to the optimal classifier $\psi_{\theta_r}$ as explained in the first paragraph in Section 3.1.

**Lemma 2** *Define* $G(x', \pi(\theta)) = (1 - cK(x', \pi(\theta)))K(x', \pi(\theta))$, $0 < c \le 1$. $G(x', \pi(\theta))$ *is a concave function of* $\pi(\theta)$.

It is important to choose a weighting scheme that renders a concave function $G$ as it guarantees the acquisition function to be larger than or equal to 0, so that adding a new observation helps to reduce weighted MOCU to effectively guide active learning.

**Lemma 3** $\forall x \in \mathcal{X}, U^w(x; \pi(\theta)) \ge 0$.

**Lemma 4** *At the $n$-th active learning iteration, if* $U^w(x; \pi^n(\theta)) = 0$, $\forall x \in \mathcal{X}$, $\mathcal{M}^w(\pi^n(\theta)) = 0$.

This lemma states that if the acquisition function values of all candidates with respect to $\pi(\theta)$ are 0, the weighted MOCU is 0. By Lemma 1, so is MOCU. With these, we can conclude that the OBC with respect to $\pi(\theta)$ has converged to the optimal classifier. This is significant when comparing with original ELR and MOCU-based methods as we have shown that this is not the case for them, which may get stuck earlier and therefore lose the long-term efficiency.

**Lemma 5** *If following some policy a candidate $x$ is measured infinitely often almost surely, then* $\lim_{n \to \infty} U^w(x; \pi^n(\theta)) = 0$ *almost surely.*

Intuitively, if a candidate has been measured many times, there is no benefit to measure it again.

With these lemmas, we can prove the convergence of weighted-MOCU based active learning:

**Theorem 1** *Assume that both $\mathcal{X}$ and $\Theta$ are discrete with finite elements, the true model parameter $\theta_r \in \Theta$ and the prior distribution $\pi^0(\theta)$ over $\Theta$ satisfies $\pi^0(\theta_r) > 0$; then for the active learning algorithm defined by the acquisition function (10), we have* $\lim_{n \to \infty} \mathcal{M}(\pi^n(\theta)) = 0$ *almost surely.*

**Proof.** As the number of active learning iterations $n \to \infty$, following the acquisition function (10), some of the candidates can be measured infinite times. Define $\mathcal{X}_A \subset \mathcal{X}$ as the set whose candidates have been measured infinite times. Denote the measuring sequence of the candidates following (10) as $\{x_n\}$, we have: $\exists N$, $s.t.$ $\forall n > N, x_n \in \mathcal{X}_A$. Based on Lemma 5, $\lim_{n \to \infty} U^w(x_n; \pi^n(\theta)) = 0$.

On the other hand, since with the weighted MOCU $U^w(x_n; \pi^n(\theta)) = \max_{x \in \mathcal{X}} U^w(x; \pi^n(\theta))$, then $\lim_{n \to \infty} U^w(x_n; \pi^n(\theta)) = 0$ indicates that $\forall x \in \mathcal{X}$, $U^w(x; \pi^n(\theta))$ uniformly converges to 0. Based on Lemma 4, $\lim_{n \to \infty} \mathcal{M}_n^w = 0$ and we can conclude the proof with Lemma 1.

Figure 3: The expected OBC error regret comparison between different active learning algorithms on binary classification.

## 4 EMPIRICAL RESULTS

We benchmark our weighted-MOCU method with other active learning algorithms, including random sampling, MES (Sebastiani & Wynn, 2000), BALD (Houlsby et al., 2011) and ELR (Roy & McCallum, 2001), on both simulated and real-world classification datasets. In the following experiments, we set $c = 1$ for the weighted MOCU function. The code for our experiments is made available at `https://github.com/QianLab/WMOCU_AL`.

**Simulated experiments.** In addition to the one-dimensional simulated example introduced in Section 1, we test our model on a similar simulation setting as the *block in the middle* dataset in (Houlsby et al., 2011), where noisy observations with flip error are simulated in a block region on the decision boundary. We generate data based on a two-dimensional Bayesian logistic regression model: $p(y = 1|\boldsymbol{x}, \boldsymbol{w}, b) = \frac{1}{1 + \exp(-\boldsymbol{w}^T \boldsymbol{x} - b)}$ with $\boldsymbol{x} \in [-4, 4]^2$. The block region is within $[-0.5, 0.5]^2$ with the flip error rate equal to 0.3. For the model parameter prior, $w_1 \sim \mathcal{U}(0.3, 0.8)$ is uniformly distributed and $w_2 \sim \mathcal{U}(-0.25, 0.25)$ and $b \sim \mathcal{U}(-0.25, 0.25)$; $w_1, w_2$ and $b$ are independent.

We randomly sample 100 particles from the parameter prior with one of the particles as the true model parameter. The five active learning algorithms are compared for 500 iterations by the OBC

error with respect to the testing data generated from the true model. We repeat the simulations for 500 runs and plot the average performance with standard deviation bars in Fig. 3. The error regret is defined as the error difference between the OBC and the true optimal classifier. From the figure,

MES simply chooses the candidates with the predictive probability closest to 0.5, it can sample many noisy observations from the block region. ELR performs well in the first several iterations but poorly after 200 samples. Our weighted MOCU performs the best.

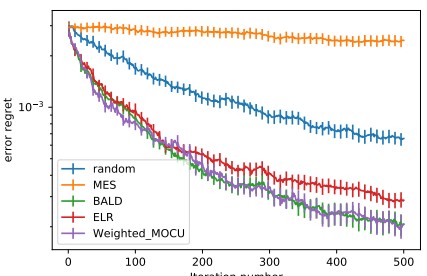

Figure 4: The expected OBC error regret comparison between different active learning algorithms on 3 class classification.

We have also benchmarked our weighted-MOCU based method with other active learning methods for a synthetic multi-class classification problem. We assume that the probabilistic model $p(y|\boldsymbol{x},\sigma_y^2) = f_y(\boldsymbol{x},\sigma_y^2)/\sum_{y'} f(\boldsymbol{x},\sigma_{y'}^2)$ with $\boldsymbol{x} \in [-2,2]^2$, $y \in \{0,1,2\}$ and $f_y = \exp(-(x - m_y)^2/2\sigma_y^2)$. We set $m_y$ to be $(0,0)$, $(1,0)$, $(0,1)$ for $y = 0,1,2$ respectively; and $\sigma_y^2 \sim \mathcal{U}(1,5)$ being the uncertain parameters. Same as the previous binary classification experiment, we test for 300 runs and plot the average performance with standard deviations in Fig. 4. We can observe that ELR performs poorly in the long run while our Weighted MOCU has better empirical performance on par with BALD More results and discussion are in Appendix D & E.

**Real-world benchmark experiments.** We also present the results on the UCI User Knowledge dataset (Kahraman et al., 2013). The dataset includes 403 samples assigned to 4 classes (High, Medium, Low, Very Low) with each sample having five features in $[0,1]^5$. We have grouped the samples into two classes with 224 samples in High or Medium, 179 in Low or Very Low. We consider the first and fifth features for classification and equally divide the feature space into $4 \times 4$ bins. For the $i$-th bin, the probability of candidates belonging to High or Medium is denoted by $\theta_i, 1 \leq i \leq 16$ and $\theta_i$'s are independent and $\theta_i \sim \text{Beta}(\alpha_i,\beta_i)$, with hyper-

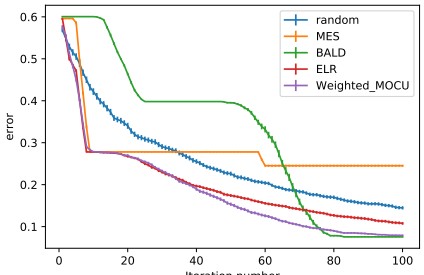

Figure 5: Classification error rate comparison on UCI User Knowledge dataset

parameters $\alpha_i$ and $\beta_i$. We present the results with the uncertainty class by setting $\alpha_i = \beta_i = 10$ in eight randomly chosen bins and for the other bins, $\alpha_i = 5, \beta_i = 2$ if the true frequency of High or Medium in the $i$-th bin is lower than 0.5 and $\alpha_i = 2, \beta_i = 5$ otherwise. We have randomly drawn 150 samples from each class as the candidate pool and perform the five different active learning algorithms. We repeat the whole procedure 150 times and the average error rates are shown in Fig. 5. While ELR clearly gets stuck in this setup, our Weighted MOCU method can converge to the optimal classifier with less samples than all the competing methods. BALD performs poorly as the bins with $\alpha = \beta = 10$ have less uncertainty but have more impact on OBC prediction and BALD fails to identify that. More comprehensive results and discussion, including results on the UCI Letter Recognition dataset (Dua & Graff, 2017), can be found in Appendix F.

## 5 CONCLUSIONS

We have identified potential convergence problems of existing ELR methods and proposed a novel active learning strategy for classification based on weighted MOCU. Our weighted MOCU directly targets at decreasing the classification error and ignores uncertainty irrelevant to the classification performance. More critically, it can capture continuous change in objective-relevant uncertainty. Hence, our new active learning can be efficient both at the beginning and in the long run with the guarantee of converging to the optimal classifier. Empirical results have demonstrated active learning guided by weighted MOCU leads to sample-efficient learning. Future work includes theoretical analysis of MOCU-guided active learning for multi-class classification, as well as developing optimization methods for active learning in continuous space.

## ACKNOWLEDGMENTS

X. Qian was supported in part by the National Science Foundation (NSF) Awards 1553281, 1812641, 1835690, and 1934904. B.-J. Yoon was supported in part by the NSF Award 1835690. The work of E. R. Dougherty and F. J. Alexander was supported by the U.S. Department of Energy, Office of Science, Office of Advanced Scientific Computing Research, Mathematical Multifaceted Integrated Capability Centers program under Award DE-SC0019303.

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

# Uncertainty-aware Active Learning for OBC:
# Appendix

In this appendix, we provide proofs of the lemmas, the pseudo-code of active learning algorithms, as well as more detailed descriptions of our experiments, additional results, and discussions on the weighted MOCU for multi-class classification.

## A. PROOFS OF LEMMAS

**Proof of Lemma 1.** Based on (3), since $C_\theta(\psi_{\pi(\theta)}, x') - C_\theta(\psi_\theta, x') \geq 0$, so $\mathcal{M}(\pi(\theta)) = 0$ iff $C_\theta(\psi_{\pi(\theta)}, x') - C_\theta(\psi_\theta, x') = 0 \ \forall x' \in \mathcal{X}, \ \forall \theta \in \text{supp}(\pi)$. In addition, in (9), $w(\pi(\theta), x', \theta) > 0$, so $C_\theta(\psi_{\pi(\theta)}, x') - C_\theta(\psi_\theta, x') = 0 \ \forall x' \in \mathcal{X}, \ \forall \theta \in \text{supp}(\pi)$ iff $\mathcal{M}^w(\pi(\theta)) = 0$, which concludes the proof.

**Proof of Lemma 2.** In the following proof, we omit the argument $x'$ in $G$ and $K$ for simplicity. Owing to the concavity of the $\min$ operator, $\min_{y'} \mathbb{E}_{\pi(\theta)}[1 - p(y'|x', \theta)]$ is a concave function of $\pi(\theta)$. With $\mathbb{E}_{\pi(\theta)}[\min_{y'}(1 - p(y'|x', \theta)]$ being a linear function of $\pi(\theta)$, based on (13), $K(\pi(\theta))$ equals to a concave function subtracting a linear function and thus is also a concave function.

As analyzed in Section 3.2, $0 \leq K(\pi(\theta)) \leq 0.5$. We define $T(\kappa) = (1 - c\kappa)\kappa, \kappa \in [0, 0.5]$, a strictly increasing and strictly concave function with $0 < c \leq 1$. $G(\pi(\theta)) = T[K(\pi(\theta))]$ is a composite function of $T$ and $K$. So we conclude the proof with the property of the concavity for the composite functions:

$$\begin{aligned} T[K(\lambda\pi_1(\theta) + (1-\lambda)\pi_2(\theta))] &\geq T[\lambda K(\pi_1(\theta)) + (1-\lambda)K(\pi_2(\theta))] \\ &\geq \lambda T[K(\pi_1(\theta))] + (1-\lambda)T[K(\pi_2(\theta))]. \end{aligned} \tag{16}$$

The first inequality is because $T$ is increasing and $K$ is concave; and the second inequality holds as $T$ is a concave function.

**Proof of Lemma 3.** Since $\pi(\theta) = \sum_y p(y|x)\pi(\theta|x, y)$, by Jensen's inequality, we have $G(x', \pi(\theta)) \geq \mathbb{E}_{y|x}[G(x', \pi(\theta|x, y))]$ as $G$ is a concave function. So the weighted MOCU acquisition function:

$$U^w(x; \pi(\theta)) = \mathbb{E}_{x'}[G(x', \pi(\theta))] - \mathbb{E}_{x'}[\mathbb{E}_{y|x}[G(x', \pi(\theta|x, y))]] \geq 0. \tag{17}$$

**Proof of Lemma 4.** We will prove the contrapositive of the lemma: assuming $\mathcal{M}^w(\pi^n(\theta)) > 0$, $\exists x \in \mathcal{X} \ s.t. \ U^w(x; \pi^n(\theta)) > 0$.

Based on (15), $M^w(\pi^n(\theta)) > 0$ indicating $\exists x \in \mathcal{X} \ s.t. \ K(x, \pi^n(\theta)) > 0$. It is sufficient to show that if $K(x, \pi^n(\theta)) > 0$, then $U^w(x; \pi^n(\theta)) > 0$. To prove that, we only need to prove $G(x, \pi^n(\theta)) > \mathbb{E}_{p^n(y|x)}[G(x, \pi^n(\theta|x, y))]$; then by (17), $U^w(x; \pi^n(\theta)) > 0$.

Since $G$ is a concave function, we know $G(x, \pi^n(\theta)) \geq \mathbb{E}_{p^n(y|x)}[G(x, \pi^n(\theta|x, y))]$. With $\pi^n(\theta) = \sum_y p^n(y|x)\pi^n(\theta|x, y)$, we can rewrite (16) as:

$$T[K(x, \pi^n(\theta))] \geq T[\mathbb{E}_{p^n(y|x)}[K(x, \pi^n(\theta|x, y))]] \geq \mathbb{E}_{p^n(y|x)}[T[K(x, \pi^n(\theta|x, y))]].$$

The second equality holds only if $\forall y \in \{0, 1\}, \ K(x, \pi^n(\theta|x, y)) = K(x, \pi^n(\theta))$, which means that to prove $G(x, \pi^n(\theta)) > \mathbb{E}_{y|x}[G(x, \pi^n(\theta|x, y))]$, we just need to show $\exists y \in \{0, 1\}, \ K(x, \pi^n(\theta|x, y)) \neq K(x, \pi^n(\theta))$. In the following proof, we will show if $K(x, \pi^n(\theta)) > 0$, then $\exists y \in \{0, 1\}, \ s.t. \ K(x, \pi^n(\theta|x, y)) \neq K(x, \pi^n(\theta))$.

Denote $\hat{y} = \arg\max_y p^n(y|x)$. By (14) we have:

$$K(x, \pi^n(\theta)) = \sum_{\theta \in \text{supp}(\pi^n)} \pi^n(\theta)[\max_y p(y|x, \theta) - p(\hat{y}|x, \theta)]. \tag{18}$$

Since $K(x, \pi^n(\theta)) > 0$, the parameter set $\Theta_o^n = \{\theta \in \text{supp}(\pi^n) : \arg\max_y p(y|x, \theta) \neq \hat{y}\}$ is not empty. We only keep the nonzero terms in $K$:

$$K(x, \pi^n(\theta)) = \sum_{\theta \in \Theta_o} \pi^n(\theta)[\max_y p(y|x, \theta) - p(\hat{y}|x, \theta)]. \tag{19}$$

For binary classification, $\hat{y} = \arg\max_y p^n(y|x)$, indicating that the predictive probability $p^n(\hat{y}|x) \geq 0.5$. For $\theta \in \Theta_o$, $p(\hat{y}|x, \theta) < 0.5$, we have: if $\theta \in \Theta_o$, $\pi^n(\theta|x, \hat{y}) = \frac{\pi^n(\theta)p(\hat{y}|x,\theta)}{p^n(\hat{y}|x)} < \pi^n(\theta)$.

If we observe $(x, \hat{y})$ in $(n+1)$-th iteration, the updated posterior predictive probability $p^n(\hat{y}|x, \{x, \hat{y}\}) \geq p^n(\hat{y}|x) \geq 0.5$ and therefore $\max_y p^n(y|x, \{x, \hat{y}\}) = \hat{y}$. Hence,

$$K(x, \pi^n(\theta|x, \hat{y})) = \sum_{\theta \in \Theta_o} \pi^n(\theta|x, \hat{y})[\max_y p(y|x, \theta) - p(\hat{y}|x, \theta)] < K(x, \pi^n(\theta)). \quad (20)$$

Since $K(\pi^n(\theta|x, \hat{y}), x) \neq K(\pi^n(\theta), x)$, we have $G(x, \pi^n(\theta)) > \mathbb{E}_{p^n(y|x)}[G(x, \pi^n(\theta|x, y))]$ and

$$\begin{aligned} U^w(x; \pi^n(\theta)) &= \mathbb{E}_{p(x')}[G(x', \pi^n(\theta))] - \mathbb{E}_{p(x')}[\mathbb{E}_{p^n(y|x)}[G(x', \pi^n(\theta|x, y))]] \\ &\geq p(x)[G(x, \pi^n(\theta)) - \mathbb{E}_{p^n(y|x)}[G(x, \pi^n(\theta|x, y))]] > 0. \end{aligned} \quad (21)$$

This concludes our proof.

**Proof of Lemma 5.** Adding a new data point $(x, y)$ to $D$, the posterior change is: $\pi^n(\theta|x, y) = \frac{\pi^n(\theta)p(y|x,\theta)}{p^n(y|x)}$. Define $\Theta_x = \{\theta \in \Theta : p(y|x, \theta) = p(y|x, \theta_r)\}$. Denote $N_x(n)$ as the times of the candidate $x$ being queried at the $n$-th iteration. Based on the posterior consistency theory we have $\sum_{\theta \in \Theta_x} \pi^n(\theta) \xrightarrow{a.s.} 1$ as $N_x(n) \to \infty$ (Gelman et al., 2013). Since $p^n(y|x) = \sum_{\theta \in \Theta} \pi^n(\theta)p(y|x, \theta)$, we have $\lim_{n\to\infty} p^n(y|x) \xrightarrow{a.s.} p(y|x, \theta_r)$. Hence $\lim_{n\to\infty} \pi^n(\theta|x, y) - \pi^n(\theta) = 0$ almost surely, which indicates $\lim_{n\to\infty} U^w(x; \pi^n(\theta)) = 0$ almost surely.

## B. WEIGHTED-MOCU BASED ACTIVE LEARNING & COMPUTATIONAL COMPLEXITY

The pseudo-code of the general active learning procedure is provided in Algorithm 2. The function ACQUISITIONFUN can be acquisition functions of various methods, including weighted-MOCU, ELR, BALD, etc.

**Computational complexity** We study the complexity of the complete active learning procedure. As we analyzed in the main text for the computation of the weighted MOCU acquisition function, the WMOCU function is called for $O(N_x N_\theta)$ times. In ACQUISITIONFUN, WMOCU is called for constant times. Finally, in the main procedure, in each iteration, ACQUISITIONFUN is called for each $x$. Hence, the total complexity of Weighted MOCU-based active learning is $O(TN_x^2 N_\theta)$.

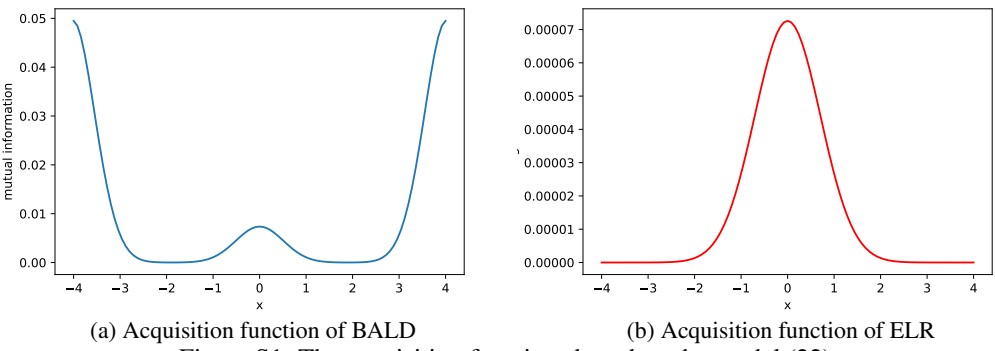

(a) Acquisition function of BALD        (b) Acquisition function of ELR

Figure S1. The acquisition functions based on the model (22).

## C. DETAILS OF THE ONE-DIMENSIONAL ACTIVE LEARNING EXAMPLE IN INTRODUCTION

The example in the Introduction of the main text is a binary classification problem with $y \in \{0, 1\}$ based on only one feature $x \in [-4, 4]$. The underlying discriminative model is based on:

$$p_c(y = 1|x, a, b) = S(x) + \epsilon(x, a, b)$$

$$S(x) = 0.6\frac{\exp(x)}{1 + \exp(x)} + 0.2$$

$$\epsilon(x, a, b) = a\exp(-x^2) + b[\exp(-(x-4)^2) + \exp(-(x+4)^2)], \quad (22)$$

---

**Algorithm 2** General active learning procedure

---

1: **function** MAINPROCEDURE( )
2:     Set a discrete candidate set $\mathcal{X}$, the probability array $p_x$, and iteration number T
3:     Set the discrete parameter set $\Theta$ and the corresponding probability array $\pi_\theta$
4:     Initialize the data set $D = \emptyset$
5:     $\pi_{\theta|D} = \pi_\theta$
6:     **for** $t = 1$ to $T$ **do**
7:         **for** $x$ in $\mathcal{X}$ **do**
8:             Store ACQUISITIONFUN$(x, \pi_{\theta|D})$ to the array $U_{\mathcal{X}}$
9:         **end for**
10:        Optimize $U_{\mathcal{X}}$ and find the maximum point $x^*$
11:        Obtain the label $y^*$ corresponds to $x^*$ and update $D = D \cup \{x^*, y^*\}$
12:        **for** $\theta$ in $\Theta$ **do**
13:            Update $\pi_{\theta|D} \propto \pi_{\theta|D} \cdot p(y^*|x^*, \theta)$
14:        **end for**
15:        $\mathcal{X} = \mathcal{X}/\{x^*\}$
16:     **end for**
17: **end function**

---

where $\boldsymbol{\theta} = (a, b)^{\mathrm{T}}$ is the uncertain parameter vector, with $a$ and $b$ independently uniformly distributed on the intervals $[-0.1, 0.1]$ and $[-0.2, 0.2]$ respectively. The discriminative model equals to a sigmoid function $S(x)$ plus the perturbation $\epsilon(x, a, b)$, a mixed Gaussian function that changes with $a$ and $b$, the uncertainty class of classifiers can be constructed by such deviations on $S(x)$. The discriminative model has higher uncertainty near $x \pm 4$, which depends on the value of $b$, than the uncertainty near $x = 0$, which depends on $a$. Value of $a$ and $b$ has negligible influence on the $p(y = 1|x)$ near $x \pm 4$ and $x = 0$, respectively. So by observing data at $x = \pm 4$, the uncertainty on $p_c(y = 1|x = 0)$ will not be reduced significantly.

In Figs. S1(a) and (b), we show how the acquisition functions of BALD and ELR change with respect to $x$. It is clear that the acquisition function of BALD at $x = \pm 4$ should have the largest value since $p(y|x)$ has the highest uncertainty. On the other hand, since $p(y|x = \pm 4)$ is always above or below 0.5, the specific value of $p(y|x, a, b)$ will not affect the corresponding optimal Bayesian classifier (OBC) and therefore the loss reduction in Fig. S1(b) at $x = \pm 4$ is always 0; the acquisition function of ELR around $x = 0$ can be larger than 0, since knowing the specific value of $p(y|x = 0, a, b)$ can reduce the classification error.

### D. DETAILS OF THE BINARY CLASSIFICATION EXAMPLE IN SECTION 3.2

In the binary classification problem, $\Theta = \{\theta_1, \theta_2\}$, $\mathcal{X} = \{x_1, x_2\}$. The probabilistic model setting for the two candidates is symmetric:

$$p(y_1|x_1, \theta_1) = (0.6, 0.4), p(y_1|x_1, \theta_2) = (0.3, 0.7)$$
$$p(y_2|x_2, \theta_1) = (0.7, 0.3), p(y_2|x_2, \theta_2) = (0.4, 0.6)$$

There are three intervals corresponding to the linear function pieces of MOCU in Fig. 2: $[0, 0.33]$, $(0.33, 0.67]$ and $(0.67, 1]$. In the three intervals, $\psi_{\pi(\theta)}(x_1)$, the OBC predictions of $x_1$ are 1, 1 and 0, respectively; $\psi_{\pi(\theta)}(x_2)$, the OBC predictions of $x_2$ are 1, 0, and 0, respectively.

In Fig. 2 we set the prior $\tilde{\pi}(\theta_1) = 0.15$, then based on the Bayes's rule we can obtain the posterior with the observations of $(x_1, y_1)$. Based on the observation result of $y_1$, the posteriors are $\tilde{\pi}(\theta_1|x_1, y_1 = 0) = 0.2609$ and $\tilde{\pi}(\theta_1|x_1, y_1 = 1) = 0.0916$, both of which fall into the first linear piece of MOCU.

### E. MULTI-CLASS CLASSIFICATION

Although we have shown in the main text that our weighted-MOCU can achieve good empirical performance of converging to OBC with the simulated multi-class classification experiment, active learning for multi-class classification problems can be complicated. The weighting function (10) adopted in the main text may not have the same theoretical convergence guarantee to the optimal

classifier if applied to multi-class classification problems. Here we just show a counter example, for which Lemma 4 does not hold if using the same weighting function.

Assume a three-class classification problem $y \in \{0, 1, 2\}$. The candidate pool only has one candidate $\mathcal{X} = \{x\}$ and the parameter set $\Theta = \{\theta_1, \theta_2, \theta_3\}$. In addition we set the probabilistic model $p(y|x, \theta)$ and the prior $\pi(\theta)$ as shown in Tables S1 and S2, and calculate the posterior and posterior predictive probabilities. In the tables, $y_o$ denotes the one-step-look-ahead observation corresponding to $x$, and $x$ is omitted for simplicity. Without loss generality, we just set the weighted MOCU parameter $c = 1$.

|  | $p(y|\theta_1)$ | $p(y|\theta_2)$ | $p(y|\theta_3)$ | $p(y)$ | $p(y|y_o = 0)$ | $p(y|y_o = 1)$ | $p(y|y_o = 2)$ |
|---|---|---|---|---|---|---|---|
| $y = 0$ | 0.4 | 0.4 | 0.4 | 0.4 | 0.4 | 0.4 | 0.4 |
| $y = 1$ | 0.3 | 0.1 | 0.5 | 0.3 | 0.3 | 0.327 | 0.273 |
| $y = 2$ | 0.3 | 0.5 | 0.1 | 0.3 | 0.3 | 0.273 | 0.327 |

Table S1. The probabilities of $p(y|x, \theta)$ and $p(y|x, y_o)$.

|  | $\pi(\theta)$ | $\pi(\theta|y_o = 0)$ | $\pi(\theta|y_o = 1)$ | $\pi(\theta|y_o = 2)$ |
|---|---|---|---|---|
| $\theta = \theta_1$ | 0.8 | 0.8 | 0.8 | 0.8 |
| $\theta = \theta_2$ | 0.1 | 0.1 | 0.17 | 0.03 |
| $\theta = \theta_3$ | 0.1 | 0.1 | 0.03 | 0.17 |

Table S2. The prior and posterior of $\pi(\theta)$.

Here two properties in the setting are worth mentioning:

1. $\pi(\theta_1)$ is close to 1 and as a result $\forall y_o \in \{0, 1, 2\}$, we have $\max_y p(y) = \max_y p(y|y_o) = \max_y p(y|\theta_1) = 0.4$;

2. $p(y|\theta_2)$ and $p(y|\theta_3)$ are symmetric and $\pi(\theta_2) = \pi(\theta_3)$, as a result $\forall y_o \in \{0, 1, 2\}$, $\pi(\theta_1) = \pi(\theta_1|y_o) = 0.8$ and therefore $\mathbb{E}_{\pi(\theta)}[\max_{y'} p(y|\theta)] = \mathbb{E}_{\pi(\theta|y_o)}[\max_{y'} p(y|\theta)] = 0.8 \times 0.4 + 0.2 \times 0.5 = 0.42$.

Recall that the $K$ function and weighted MOCU are:

$$K(\pi(\theta)) = \mathbb{E}_{\pi(\theta)}[\max_{y'} p(y|\theta)] - \max_{y'} p(y), \tag{23}$$

$$\mathcal{M}^w(\pi(\theta)) = [1 - K(\pi(\theta))] \cdot K(\pi(\theta)). \tag{24}$$

Therefore, we have $\forall y_o \in \{0, 1, 2\}$, $K(\pi(\theta)) = K(\pi(\theta|y_o)) = 0.02$ and $\mathcal{M}^w(\pi(\theta)) = \mathcal{M}^w(\pi(\theta|y_o)) > 0$. On the other hand,

$$U^w(\pi(\theta)) = \mathcal{M}^w(\pi(\theta)) - \mathbb{E}_{p(y_o)}[\mathcal{M}^w(\pi(\theta|y_o))] = 0, \tag{25}$$

which means that the algorithm may get stuck. Here we just give an extreme case where only one candidate is in the search pool, but it is straightforward to build a more practical example based on what we have shown here.

We can see from the example that, unlike in the cases of binary classification problems, the weighting function $1 - cK$ may remain unchanged for a single observation in multi-class problems. Because of this, the weighted-MOCU algorithm may get stuck. Since OBC prediction is the maximum of the predictive distribution $p(y|x)$, the weight function is introduced to capture the changes of $p(y|x)$, as that indicates the potential shift of OBC prediction in the long run. $K$ is a function of $\max_y p(y|x)$, in binary case, $\max_y p(y|x)$ must change as $p(y|x)$ changes. However, in multi-class problems, the probability of the optimal label $\max_y p(y|x)$ may remain unchanged, when the probability of other labels change, just like in the example above where $\max_y p(y) = \max_y p(y|y_o = 1)$. In the next section, we propose a weighting function that can capture the change of any element in $p(y|x)$.

F. ANOTHER WEIGHTED MOCU SCHEME FOR MULTI-CLASS CLASSIFICATION

To extend the weighted MOCU scheme suit for the multi-class problem, we propose a weight function that can capture the change of $p(y|x)$. The weighting function is defined as the softmax of

$p(y|x)$:

$$w(\pi(\theta), x', \theta) = \frac{\exp(\max_y p(y|x))}{\sum_i \exp(p(y_i|x))}, \tag{26}$$

where $p(y|x)$ is the posterior predictive distribution at the current active learning iteration. We compare this Weighted MOCU with other active learning algorithms empirically on the synthetic three-class classification problem and the performance comparison is shown in Fig. S2. This new Weighted MOCU (Weighted MOCU2) performs slightly better than other algorithms on this multi-class classification problem.

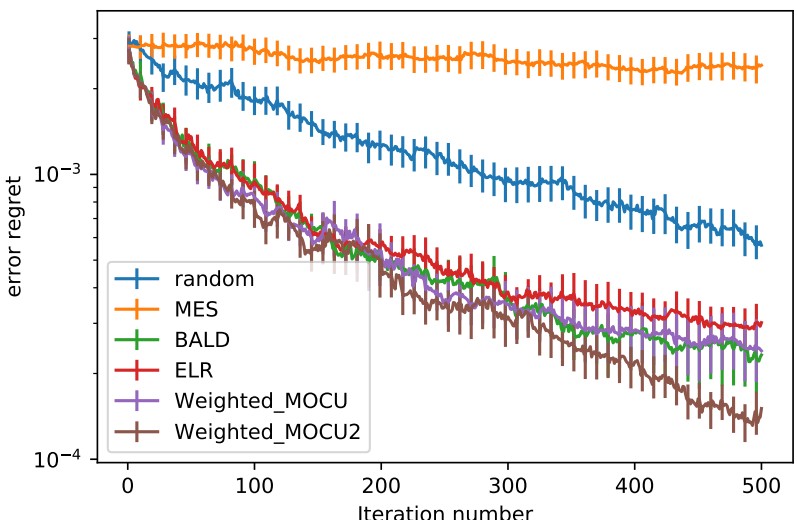

Figure S2. The expected OBC error regret comparison between different active learning algorithms for the three-class classification problem.

## G. ADDITIONAL SYNTHETIC EXPERIMENTS

We run the same synthetic experiment of Fig. 3 with a different prior setting: $w_1 \sim \mathcal{U}(0.3, 0.8)$, $w_2 \sim \mathcal{U}(-0.02, 0.02)$ and $b \sim \mathcal{U}(-0.25, 0.25)$, and the results is shown in Figure S3. The performance shows that only our Weighted MOCU method performs better than the random benchmark.

Here we benchmark different active learning strategies for OBC with another synthetic example. Assume the classification problem with two dimensional input features $\boldsymbol{x} = (x_1, x_2) \in \mathbb{R}^2$ and binary class labels $y \in \{0, 1\}$. The computational model is derived by a decision boundary in a quadratic form: $x_2 = ax_1^2 + bx_1 + c$, i.e. $p(y = 1|\boldsymbol{x}, a, b, c) = \mathbb{1}(x_2 > ax_1^2 + bx_1 + c)$. The parameter vector $\theta = (a, b, c) \in \mathbb{R}^3$ is uncertain and the true model is characterized by a true parameter $\theta^*$. Unlike Monte Carlo sampling in the main text, here we consider a discrete grid setting for both input space and parameter space with discretization for each variable as follows:
1. $x_1$ ranges in $[-0.5, 0.5]$ with increment 0.05
2. $x_2$ ranges in $[0, 2]$ with increment 0.1.
3. $a$ ranges in [-4.3, -3.8] with increment 0.05,
4. $b$ ranges in [-0.25, 0.25] with increment 0.05,
5. $c$ ranges in [1, 2] with increment 0.05.

For now, we simply assume that the distributions over the feature space and parameter space are all uniform to illustrate the effectiveness of MOCU-based active learning. With prior knowledge of the system of interest, knowledge-driven prior should be incorporated. Following the weighted-MOCU based active learning algorithm in Algorithm 1, we can sequentially query the true system and reduce the model uncertainty in a way that maximally reduces the classification error of the corresponding OBC.

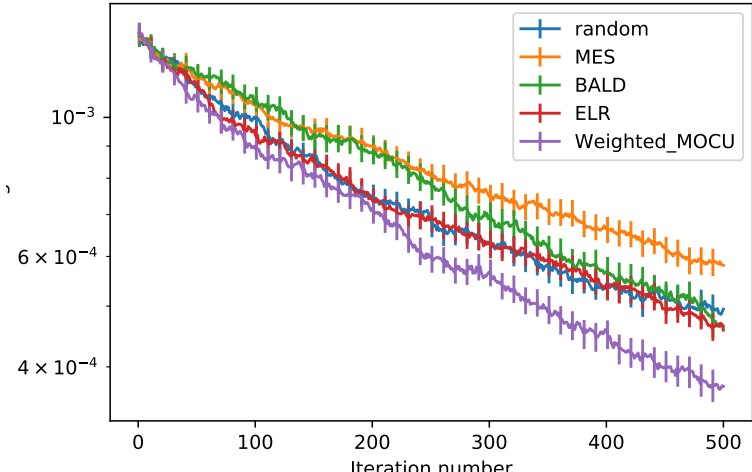

Figure S3. The expected OBC error regret comparison between different active learning algorithms on binary classification.

Now we assume that when querying the system, the class label is given with a heterogeneous random flipping error with the error probability being a function of $x_1$: $p(y = 1|z = 0) = p(y = 0|z = 1) = 0.3 \times (1 - 4x_1^2) + 0.1$. Therefore, when $x_1 = 0$, the flipping error is 0.4; and when $x_1 = \pm 0.5$, the flipping error is 0.1. We have implemented the same methods as in the main text with 50 iterations and 100 runs, The active learning results are illustrated in Fig. S4. As we can see, in this figure, MES does not perform well as it cannot differentiate between model uncertainty and observation error. ELR performs similarly to BALD and our weighted-MOCU based method at the beginning, but then it gets stuck before finding the true boundary. BALD and our weighted MOCU perform similarly. This is because in this setting $p(y|x, \theta)$ is either 1 or 0, so there is no irrelevant uncertainty with which $p(y|x, \theta)$ is always larger or smaller than 0.5 but the value is uncertain.

In addition to the average performance comparison, we deliberately choose one of the runs in which the ELR method gets stuck to better illustrate the difference between the existing ELR methods and the proposed weighted-MOCU based method. In this run, the randomly chosen parameters are $(a = -3, b = 0, c = 1.9)$. Fig. S5(a) shows the error regret (the OBC error minus the true optimal classifier error) comparison, in which ELR gets stuck and the weighted-MOCU based method reaches 0. Notice that the y-axis is in the logarithm scale, so the vertical line in the WMOCU plot implies that the value turns to 0. Error regret equals to 0 indicates that the OBC classifier equals to the true optimal classifier, but in practice we don't know the true optimal classifier, so we need the value of MOCU to quantify the expected error difference between OBC and the optimal classifier of each $\theta = (a, b, c)$. Fig. S5(b) shows the changes of MOCU value during the two active learning procedures. Not surprisingly, the MOCU value during the iterations of the ELR method also gets stuck, while the MOCU value in the iterations of the weighted-MOCU method continues to decrease. Fig. S5(c) shows the changes of the maximum value of acquisition function in each iteration. The acquisition function of ELR decrease to 0 after 22 iterations, and that explains why ELR gets stuck. On the other hand, the maximum acquisition function of WMOCU is always positive as the corresponding MOCU is positive, until it gets close to $10^{-16}$, which is the rounding error in floating point arithmetic. In theory, as the observation is noisy, we can not be sure of the optimal prediction. Therefore, the MOCU and the acquisition function of weighted-MOCU should always be positive, which is demonstrated in the figures.

We have also performed an experiment to show the algorithm performance change under different noise levels. We set the flipping error rate as $p(y \neq z|\boldsymbol{x}) = \epsilon \times (1 - 4x_1^2) + \epsilon, 0 \leq \epsilon \leq 0.25$. Therefore, when $x_1 = 0$, the flipping error is $2\epsilon$; and when $x_1 = \pm 0.5$, the flipping error is $\epsilon$. We perform the same methods with 100 iterations and 100 runs on the noise level $\epsilon = 0.05$ and $\epsilon = 0.25$. The resulting active learning performance curves are illustrated in Fig. S6. We can

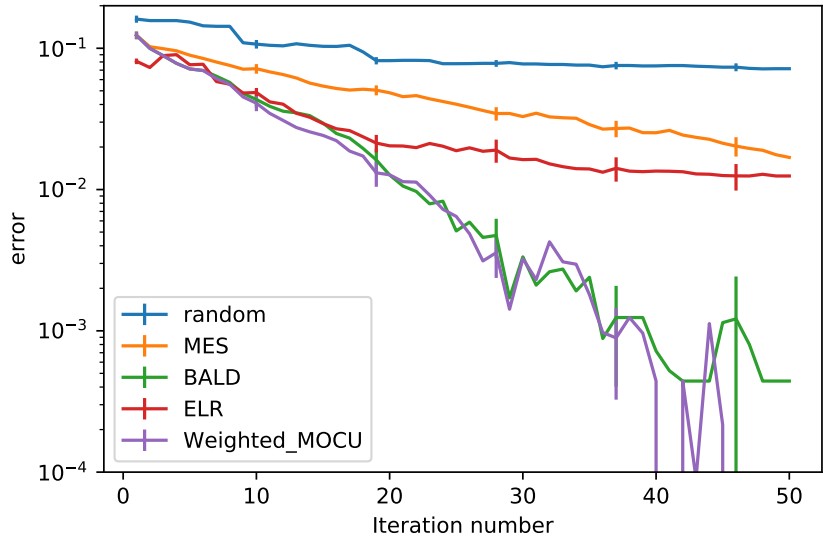

Figure S4. The expected OBC error comparison between different active learning algorithms in the setting with heterogeneous observation error.

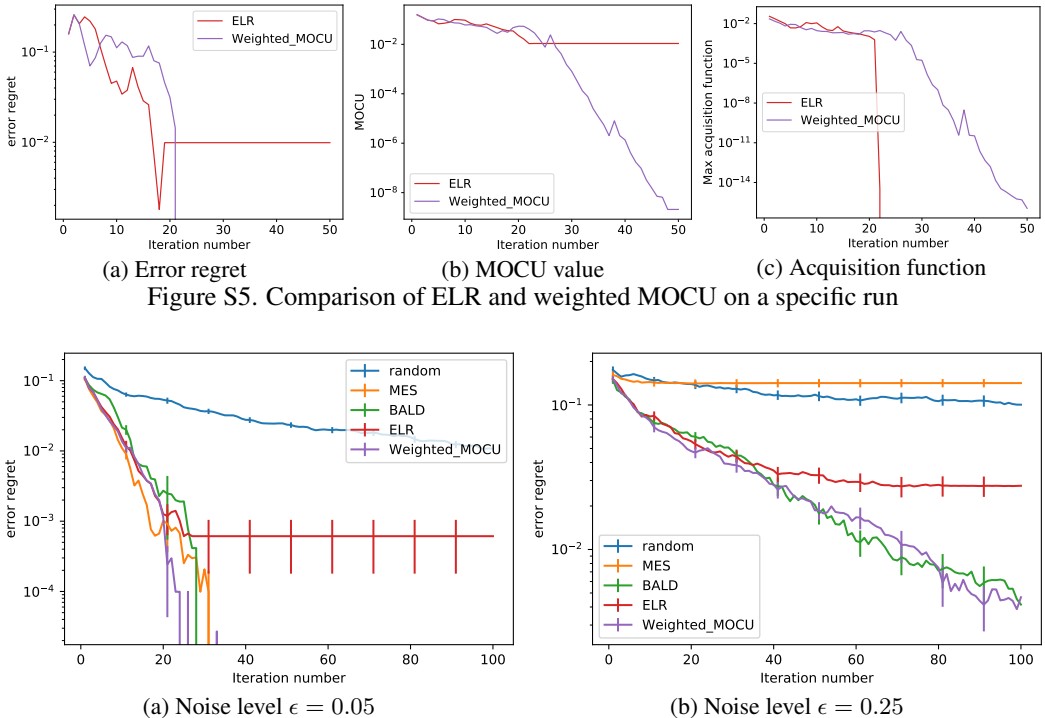

(a) Error regret  (b) MOCU value  (c) Acquisition function

Figure S5. Comparison of ELR and weighted MOCU on a specific run

(a) Noise level $\epsilon = 0.05$  (b) Noise level $\epsilon = 0.25$

Figure S6. Active learning algorithm performance comparison with different noise levels

see from the figure that the performance of MES degrades significantly with high noise while the performance of other methods does not appear to be very sensitive to the increasing noise level. .

## H. REAL-WORLD BENCHMARK EXPERIMENTS.

We here present the complete results on the UCI User Knowledge dataset (Kahraman et al., 2013). In addition to the uncertainty class setup in the main text, we have tested two other setups of hyper-parameter values: 1) 'uniform prior' with $\alpha_i = \beta_i = 1$, and 2) 'good prior' with $\alpha_i = \beta_i = 10$ in eight bins chosen randomly, for other bins $\alpha_i = 5, \beta_i = 2$ if the true frequency of High or Medium

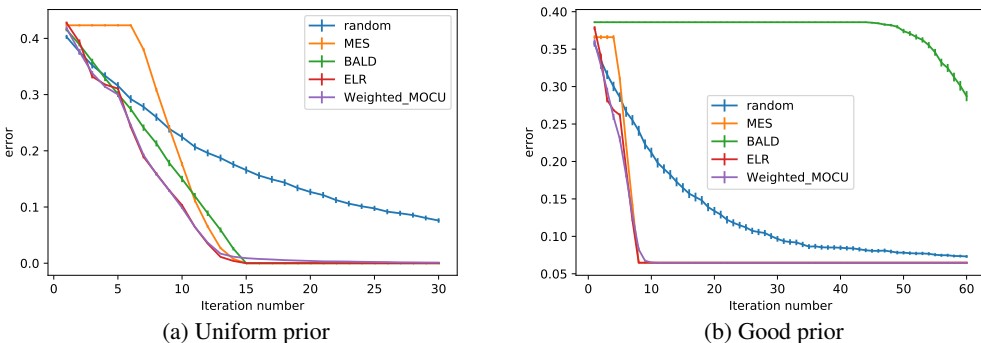

(a) Uniform prior                                 (b) Good prior

Figure S7. Classification error rate comparison on UCI User Knowledge dataset

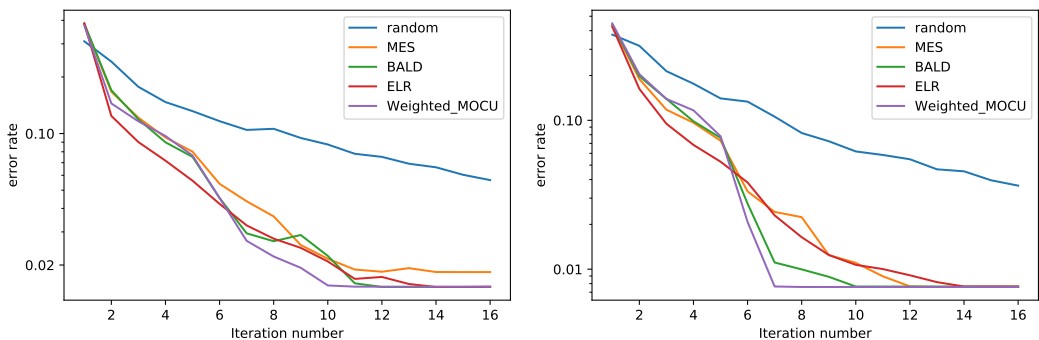

(a) Performance of letter E vs. F classification    (b) Performance of letter P vs. D classification

Figure S8. Classification error rate comparison on UCI Letter Recognition dataset

in the $i$-th bin is higher than 0.5 and $\alpha_i = 2, \beta_i = 5$ if the frequency is lower than 0.5. We also randomly draw 150 samples from each class as the candidate pool and perform the five different active learning algorithms. We repeat the whole procedure 150 times and the average error rates are shown in Fig. S7. In both Fig. S7a and Fig. S7b, ELR performs the best in these two setups while our Weighted MOCU performs similarly. BALD performs reasonably in Fig. S7a but it again performs poorly in Fig. S7b. This is because the bins with $\alpha = \beta = 10$ have less uncertainty but have more impact on OBC prediction and BALD fails to identify that in this setup again.

We also present the results on the UCI Letter Recognition dataset (Dua & Graff, 2017). Letter Recognition is a multi-class classification dataset with each sample having 16 numerical features generated from typed images of the capital letters in the English alphabet. We select two pairs of hard-to-distinguish letters: E vs. F and D vs. P. The total number of training samples is 1543 and 1608 for E vs. F and D vs. P, respectively. Active learning algorithms are applied with Bayesian logistic regression models. We randomly take 100 data points first to construct the prior, and use the rest of the data as the pool to test the five active learning algorithms. For prior construction, we train a logistic regression model on the 100 data points and take the trained parameters as the mean of a normal distributed prior with the variance equal to 1. Then we sample 1000 particles from the prior as the uncertain parameter set. We repeat the whole procedure 100 times and the average error rates are shown in Fig. S8. Unlike the synthetic datasets, the real-world datasets have no corresponding true models. We can only find the optimal models that approximate the data best. However, we can still see the trends of different algorithms. Compared with random sampling, all the algorithms quickly converge to the optimal models. ELR performs the best in the first several iterations, while converges slowly in the latter iterations. Our weighted MOCU based method is again demonstrated to converge faster than other competing methods.

It is clear from all our experiments for both simulated and real-world data that, in addition to its theoretical guarantee for active learning with OBC, our weighted MOCU method has achieved consistently better or similar empirical performance compared to the best performing ones among the

existing pool-based active learning methods, approaching the corresponding OBCs faster with fewer labeled samples.

