# OpenReview forum: "Uncertainty-aware Active Learning for Optimal Bayesian Classifier"
_ICLR.cc/2021/Conference — ICLR 2021 Poster_

### Official Review · AnonReviewer3 · 2020-10-28
**Difficult to read**

**Rating:** 5
**Confidence:** 2

**Review:**

The authors of this paper introduced a new acquisition function of active learning for optimal Bayesian classifier. The new query strategy is based on mean objective cost of uncertainty, defined as the expected difference between losses of the optimal Bayesian classifier and the optimal classifier.

I think this paper can benefit from revisions to improve clarity. Unfortunately, in the current state of writing, I found it very difficult to understand what this approach is doing exactly. The lack of clarity makes it hard to appreciate the interestingness of the proposed approach. In the following, I will list some possible improvements and my confusions.

1, In the abstract, please rewrite the sentence "To improve convergence ... classification error." This sentence is probably the most important summary of this work, but it's so long and dense that it's very difficult to parse. I believe people read abstract to get a general idea of what this paper is, not a dense summary of what the technical details are.

2, The introduction should be called related work. Especially in 2nd-3rd paragraph, the authors tried to pack all competing algorithms in and explain why they don't work well. It is too detailed, I think. I expect more high-level descriptions of why this problem is important, where the field is now, or why the authors think this is an important problem to solve rather than, e.g., solving active learning for regression.

3, The authors have a tendency of defining a symbol or an abbreviation, and expect the readers to register them in their memory. It would help the clarity significantly if the authors could just repeat in English what \pi (or \phi or C_\theta , M, U etc) is again when they are mentioned.

4, It's not clear to me why the entire introduction and definition of MOCU is under section 3.1 Analysis of ELR Methods. Perhaps section 3.1 needs to be segmented into more subsections.

5, I'd recommend that the authors leave the most important theorem in the main paper and move all the less important lemmas and proofs to appendix. Then, the authors can have more space explaining the intuitions behind the proofs and the newly designed weighted MOCU. It is nice to see the convergence analysis, but it also makes me wonder how useful it really is. The theorem is saying, as we get infinite samples, we get the optimal classifier, under a bunch of conditions. It's nice to have, but it almost feels like every algorithm that can loop through all possible inputs can do that. What about the convergence rate that we care more about? Or how many active learning iteration is needed to achieve a certain performance.

6, While it is unclear how useful the theoretical guarantees are, it is also unclear if the empirical results should enough evidence. Only toy datasets were examined, and the performance of the proposed approach is quite similar to other competitors.

---

> ### Author Response · Authors · 2020-11-23
> **Response to AnonReviewer3**
>
> 1. Revision
> We thank the reviewer for constructive critiques and have significantly revised the paper following the reviewer's suggestions. We have revised the abstract for a clearer summary. In the introduction, we have removed some descriptions of competing algorithms and emphasized that we focus on the active learning methods that directly maximize the learning performance. We have separated the MOCU introduction in a new subsection. Moreover, we have significantly revised Section 3.2 and added Fig. 2, which can provide a better intuitive illustration of MOCU and weighted MOCU differences for understanding the issues of ELR/MOCU methods. We believe it also better motivates our weighted-MOCU based active learning method development.
>
> 2. Significance of the convergence proof
> We agree that many methods can converge to the true model asymptotically, and as a result, can converge to the optimal classifier of the true model including cyclic sampling and BALD. However, these methods may not sample efficiently as they may reduce the model uncertainty that do not directly affect the classification performance as we explained.
> The significance of our proof is that: 1) ELR/MOCU based methods directly reduce the uncertainty affecting classification, but we analyzed that they may not converge due to the myopic nature. 2) Our weighted-MOCU based method approximates the ELR method and still directly reduces the uncertainty affecting classification. Therefore it is more efficient, by prioritizing queries directly improving classification, than other converging methods, including BALD, which has been shown empirically in different setups in our experiments.
>
> 3. Experiments are only on toy datasets.
> We would like to emphasize our paper focuses on theoretical results to make sure that our weighted-MOCU active learning can achieve better data efficiency than existing methods. With the datasets in different setups, including the real-world UCI datasets, we have validated our theoretical results empirically.  Specifically, we focus on the performance comparison between ELR and weighted MOCU to verify the theorem. We also have tried to show that other methods (MES and BALD) that reduce the total model uncertainty (instead of only the objective uncertainty that actually affects learning objectives) can perform poorly.  We note that these active learning methods can be implemented to more complicated datasets and models but demonstrating the expected convergence can take time. We will implement these algorithms for other datasets and evaluate them accordingly.

---

### Official Review · AnonReviewer4 · 2020-10-28
**Review for the paper**

**Rating:** 6
**Confidence:** 4

**Review:**


Summary:

This paper provides an interesting algorithm to address the previous Bayesian active learning query strategy in (binary) classification. By the simple modification, the algorithm can overcome the drawbacks of ELR in the convergence to the optimal classifier parameterized by $\theta_r$. In experiments, the proposed algorithm can achieve the advantages of ELR and BALD simultaneously.

Reasons for score:
Overall, I vote for the marginally above the acceptance threshold. The proposed methodology is impressive and well-motivated. However, there are some lacks in addressing the problem of ELR in a qualitative manner. The assumptions of theorems look strong. I feel that there can be room to be improved in this paper.

Pros:
1. This paper was well motivated by the drawbacks of ELR and insightful comparison of BALD and ELR. In Bayesian approaches, these issues can be interesting and valuable.
2. The proposed algorithm is simple and addresses the problem caused by only mitigating the mean difference. The proposed algorithm can diminish the mean difference and ensure that $M^w ( \pi^*(\theta) )$ converges to 0.
3. The proposed algorithm can dominate the random and ELR. Also, the prior can be used to provide better results.

Cons:
1. The problem of ELR is not verified thoroughly. The stuck in the convergence of ELR can be due to the lack of considering the long term effects. However, the detail of this phenomenon is not verified in a more detailed manner. Can you show the details of what happened in the ELR active learning? At least, I want to see the values of $U$ and $M$ when the ELR is used.
2. The proofs assume that the supports of $\pi(\theta)$ and $x$ are finite, respectively, and prior is limited to the discrete-type probability measure. These assumptions can be a good starting point. However, it is better to provide any clue that we can extend this result to more general settings.
3. The counter-example of the proposed algorithm for multi-class in the Appendix shows this paper's prematurity in multi-class problems, and there are no details to address this problem. If you can provide some clues to address the multi-class problem, it is very helpful.

Minor Comments:

1. It is not easy to follow the proofs in Section 3.1. The authors claim that the lower bound of OBC error will be canceled in the (5). The equations after (5) can imply this cancelation. However, there is no direct wording to conclude this cancelation.
2. In the equation of $\sum_y p^*(y | x) \pi^*( \theta | x, y) = \pi^*( \theta )$, $\pi^* ( \theta ) = \pi^*(\theta | x),$ it is better to clarify that $\pi^*$ is not affected by $x$.
3. In the proofs of theorem 1,  the notation of $X_A (w)$ is not consistent with the previous notation of  $X_A$.
4. The infinite querying for a fixed $x$ cannot be realistic in some cases. Therefore, the proof should be extended for the case that the support of $x$ is an open subset of $\mathbb{R}^p$ where $p$ is the dimension of $x.$

---

> ### Author Response · Authors · 2020-11-23
> **Response to AnonRevierwer4**
>
> Thanks very much for your comments
> 1. Details of what happened in the ELR active learning
> We have added an additional experiment in our Appendix G to compare the ELR and weighted MOCU methods. In Fig. S5, we have shown the value changes of MOCU and the maximum of the acquisition function during the active learning procedures.  In the figure, we can see in ELR active learning, after 22 iterations, the maximum of the acquisition function turns to 0 while the MOCU gets stuck on a positive value. Therefore, ELR gets stuck and is degenerated to random sampling based on the adopted tie-breaking strategy.  On the other hand, in our weighted-MOCU based method, the MOCU is positive all the time and keeps decreasing and the maximum of the acquisition function is also positive all the time. Therefore, it can query the candidates effectively both at the beginning of the active learning procedure and in the long run.
> We have also added Fig. 2 in the revision that can provide further intuition on why ELR/MOCU-based method may get stuck.  We also have explained how the strict concavity forced by our weighted MOCU can efficiently guide active learning to approach to the true optimal classifier in the long run.
>
> 2. The limit of proof
> We believe that we can extend the proof to the cases where the support of $\theta$ is continuous. In that case, $\pi(\theta)$ is a distribution, and then $K$ and $G$ are both functionals. Lemma 2 should still be valid.
> While for the cases where the support of $x$ is continuous, the proof can be difficult. However, we'd like to emphasize that our paper focuses on the pool-based active learning as we stated throughout the paper, where the support of $x$ is indeed discrete.
>
> 3. Extension for multi-class classification problem
> We have included an algorithm in Appendix F to solve the multi-class classification problem with corresponding discussion. We note that we have not proved its convergence.
> Since the OBC predictions depend on the predictive distribution $p(y|x)$, we require the weighting function to capture the update of $p(y|x)$ given one single query. The current weight $(1-cK)$  only depends on $\max_y p(y|x)$, which is enough for binary classification problems  as $\max_y p(y|x)$ changes if the $p(y|x)$ changes. However, that's not true for multi-class cases as shown by the counter examples in Appendix E. So in Appendix F, we propose to use the softmax of $p(y|x)$ as the weighting function because it's concave and can capture the change of $p(y|x)$.  As we have shown in Fig. S2, the performance of this new weighted MOCU (weighted-MOCU2) is better than the original weighted MOCU on the three-class classification problem.
>
> We have clarified our proofs and fixed the incorrect notations in our revised version.

---

### Official Review · AnonReviewer1 · 2020-10-28
**A fairly good submission**

**Rating:** 7
**Confidence:** 3

**Review:**

This paper addresses the active learning paradigm in which the learner queries an oracle to obtain the class label of some inputs. Depending on the querying strategy, the learner can improve its classification model more or less efficiently.

Among other possibilities, the authors focus on a class of methods known as Expected Loss Reduction, which is optimal in one-step-ahead risk minimization. However, the authors shed light on the fact that the long-term effect of this strategy do not necessarily allow to reach an optimal classifier. They thus introduce an alternative approach that achieves this goal by focusing on loss uncertainty.

More precisely, the authors introduce the Mean Objective Cost of Uncertainty (MOCU) that captures the expected difference between the error of Bayesian optimal classifier (BOC) and the expectation (against the parameter theta posterior) of the error of the theta-best classifier. An active learning strategy can devised by looking for a new input that (roughly speaking) will cause the largest MOCU drop.

Because the strategy consists in selecting inputs that maximizes MOCU drop, MOCU will decrease as new class labels are revealed but this does not imply that MOCU will reach its minimum (zero). To achieve long run convergence of MOCU to zero (and thus get obtain the optimal classifier), the authors propose a so-called weighted strategy that solve this issue.

Finally, the authors provide fair numerical experiments on both synthetic and real datasets. The results indicates that the proposed strategy seems to provide good results in a wider range of situations as compared to SOTA.


Major Remarks :

Although the authors provide some proof that the weight function can solve the long-run convergence issue, I wish they would provide intuitions as to why their particular choice can choose inputs that will be beneficial in this regard. The chosen weight function is going in the opposite direction of MOCU. Its effects are thus hard to interpret although it is instrumental to obtain a concave functions for the proofs.

The major drawback of the paper is that the scope of the proofs is very limited. Can the authors give insights as to what would remain valid in more realistic situations in which one has class imbalance, non-uniform utility (more general loss than 0-1), continuous parameter space ? The proof (unless I am mistaken) also do not account for approximation errors incurred by replacing expectation with empirical versions.
A few comments are provided in the appendices concerning multi-class problems.
I, however, reckon that the numerical experiments are reassuring

An algorithm is provided in the appendix but some comment on complexity as compared to prior arts in the main text would be appreciated.

Minor remark :

I am bewildered by this statement : "Converging to the true model is unnecessary and inefficient for classification". Should this be understood as the myopic strategies standpoint ?

---

> ### Author Response · Authors · 2020-11-23
> **Response to  AnonReviewer1**
>
> Thanks for your review. We address the concerns as follows:
> 1. The intuition behind the weight choice.
> We have added Fig. 2 in the revision to show the comparison of the MOCU and weighted MOCU function. We believe that it can provide further intuition why the strict concavity is desired for weighted MOCU to guide active learning efficiently considering both short-term and long-term gains. We have also generalized the weighting scheme and the weight is chosen as $1-cK$, with $c$ controlling the approximation of weighted MOCU to the original MOCU. From Fig.2 we can see with $0<c\leq 1$, the weighted-MOCU function is below the original MOCU and is strictly concave.
> 2. The limit of proof.
> Regarding non-uniform utility, we assume that the reviewer meant that the false positive cost and the false negative cost can be different. In that case, the expression of MOCU and $K$ will change, but they are still piece-wise linear functions of the  $\pi(\theta)$, and we should still choose the weight as $1-cK$. Only this time the feasible range of $c$ to have the weighted MOCU to be concave will change, depending on the utility function. As a result, our weighted-MOCU based active learning still has the theoretical convergence guarantee while ELR may still get stuck.
> With continuous parameters, $\pi(\theta)$ is a distribution, and then $K$ and $G$ are both functionals. We believe our proof should still be valid.
> 3. Complexity comparison.
> Assume all competing algorithms are applied on the same setup of pool-based active learning with $N_x$ candidates and $N_{\theta}$ parameters. ELR has the complexity of $O(TN_x^2N_{\theta})$, same as our weighted MOCU method. BALD and MES have a complexity of $O(TN_xN_{\theta})$, which is the complexity of computing the posterior.  In our original submission, the complexity analysis was in Appendix A. We have reorganized the submission with complexity analyses in both Section 3 and Appendix B in the revised version.
> 4. The confusing statement.
> That statement emphasizes the difference between converging to the true model and converging to the true optimal classifier. We have tried to revise that statement in the revised version for clearer presentation.

---

### Official Review · AnonReviewer2 · 2020-11-07
**Official Blind Review #2**

**Rating:** 6
**Confidence:** 4

**Review:**

This paper studies the label solicitation strategy in active learning. In particular, it focuses on the expected loss reduction (ELR) strategy, analyzes its problem, and modifies the original ELR method to make sure the active learner converges to the optimal classifier along learning iterations. The paper provides theoretical guarantees on the new method’s convergence. In the experiment, the proposed method is evaluated on synthetic data and UCI data. The improvement margin over the existing method is very limited.

Strong point:
1. The paper’s finding on the existing ELR method is interesting and novel.
2. The theoretical analysis of the convergence of the proposed method seems to be sound.

Weak point:
1. The experiment is conducted on low-dimensional data and the proposed method’s performance is not very competitive.
2. The notation in this paper can be confusing to readers, especially the use of (*). Star usually means the “optimal”.
3. The main paper does not have an algorithm.
4. There is no validation in experiments for the theory. In the synthetic experiment, it should be possible to simulate a case with ground truth optimal classifier and verify whether the proposed method actually converges to the optimal.

My major concern is the practical impact of the proposed method. Therefore, I recommend a weak reject for this paper (5).

Additional questions and suggestions:

1. I think the paper would be improved if there is a discussion on how the proposed method can be extended to deal with high-dimensional data and/or using deep learning models.

2. It seems to me the proposed weighted method is not the only way to guarantee convergence. But I am not sure about that. It would be nice to have some discussion about that.

3. The one-step-look-ahead strategy involving expectation model change or expected loss reduction usually suffers from the large class space for computing the expectation. The experiments are mostly conducted in a small class space. It would also be good to have a discussion about the complexity in terms of class space.

4. MES usually suffers a lot from noise (experiments in the appendix also show that). ELR methods are usually more robust to noise. I was wondering whether different noise level has been tried and how the proposed method compare with ELR on that. A similar question is also, for certain data there is a larger gap between the proposed method and ELR (in appendix). What would be the reason?

5. The visual presentation can be improved for the paper, as well as the explanations. In figure 1’s explanation, I got very confused by the “side”. What does that mean?

6. The results should be shown with error bars if experiments are conducted multiple times.

================
Update after rebuttal:

I increased the score to 6 and appreciated the revision of the paper. The readability is improved. However, I also have different opinions with the authors in terms of how empirical evaluation of algorithms should be regarded in active learning research. So I would further encourage the authors to apply their method on high-dimensional large scale data, even it may take a lot of computing resources or require actual sample acquisition.

I agree that the goal of active learning is to reduce the burden of labeling data. But it does not conflict with the requirement of dealing with high-dimensional (feature space) data. Also, I see a lot of active learning works focusing on theoretical analysis but cannot be easily put into real-world applications, which actually undermines the significance of the theory to some extent. In the real world, a lot of assumptions would be violated. As the authors also mentioned that, it is "expected" that different feature space and data quality affects the performance. Therefore, I think the theory does not spare us from justifying our methods in practice.

Last but not least, actual sample acquisition is not unrealistic if given real-world problems. So I encourage the authors to further demonstrate the nice properties of the proposed algorithms in more realistic settings in the future.

---

> ### Author Response · Authors · 2020-11-23
> **Response to AnonReviewer2 Part 1**
>
> Thank you for your comments and helpful suggestions. We address your specific questions and concerns below.
>
> Q1. Need more experiments in high-dimensional space and more complex models, such as neural networks. Performance improvement is limit.
>
> A1. (1) We would like to emphasize that the settings of pool-based Bayesian active learning are to address the lack of labeled data. The presented work aims to develop a label-efficient active learning method that provides both short-term and long-term improvements (in terms of prediction performance per candidate training sample to query), importantly with a strong theoretical guarantee of convergence. This theoretical guarantee is extremely important when the feature dimension and model complexity increases, as it ensures that our active learning scheme will converge to the true optimal classifier. Our complexity analyses show that the time complexity increases linearly with the sizes of both feature and output spaces. The main purpose of the presented experimental results is to validate our theory with empirical demonstration of the issues of the existing active learning methods and the convergence of our weighted-MOCU based method. With increasing feature dimension and model complexity, it can take much more computing hours to validate the convergence of active learning algorithms to the optimal classifier, and this is precisely why a theoretical guarantee is extremely important – as it provides confidence that the algorithm will converge to the optimal classifier, without having to empirically show it based on significant computations (or actual sample acquisitions, which is unrealistic).
>
> (2) Regarding the application to neural networks, as we focus on Bayesian classifiers here, we would need to implement Bayesian neural networks to test these methods, for which training can further add more computational burden. Hence, we leave these practical applications for future work but focus on fundamental theoretical contributions in this submission, along with empirical demonstration of its importance. Last but not least, we would like to emphasize that our weighted MOCU has the same complexity as ELR, and our method is also practical as demonstrated by previous papers with ELR methods.
>
> (3) Regarding performance improvement, in this paper, the performance comparison between ELR and weighted MOCU have verified the theorem showing that our weighted-MOCU based method achieves data efficiency both at the beginning of the active learning procedure and in the long run, by focusing only on the uncertainty that directly affects classification – instead of the total uncertainty by BALD. The experimental results in different setups have shown consistent improvements over existing methods. Of course, the improvement is dependent on the underlying feature-label distributions and data quality and therefore in some setups, the empirical improvement can be limited. However, the proposed method has clear advantage over other methods (e.g., MES and BALD) too, which may perform poorly for complex models in which some model uncertainty may not directly affect the learning objective of interest.
>
> Q2. Notation
> A2. In the revised paper, we have removed the (*) with simplified but consistent notations.
>
> Q3. Algorithm pseudocode
> A3. We have added the pseudocode, together with computational complexity analysis, of calculating the acquisition function to the main text in our revision.
>
> Q4. Validation of the theory.
> A4. We have added an additional experiment in Appendix G to compare the ELR and weighted MOCU methods for classification with noisy observations. With noisy observations, we are not sure of the optimal prediction with finite observations. Therefore, the MOCU value during the learning procedure should always be positive. In Fig. S5, we have shown the value changes of MOCU and the maximum of the acquisition function during the active learning procedures. In contrast to the ELR method getting stuck, in our weighted MOCU method, the MOCU is positive and continues to decrease during the whole procedure; and the maximum of the acquisition function is also positive, indicating the learning procedure does not get stuck. The MOCU value reaching a very small value ($10^{-8}$) at the end can validate the theory of our asymptotic convergence.  Fully verifying the asymptotical property is impossible, once again, this is exactly why the theoretical convergence proof is very important.

---

> > ### Author Response · Authors · 2020-11-23
> > **Response to AnonReviewer2 Part 2**
> >
> > Q5. Discussion on other converging method.
> >
> > A5. (1) We agree that many methods can converge to the true model asymptotically, and as a result, can converge to the optimal classifier of the true model, including cyclic sampling and BALD. However, these methods may not sample efficiently as they may reduce the model uncertainty that do not affect classification performance as we explained.
> >
> > (2) The significance of our proof is that: 1) ELR/MOCU based methods directly reduce the uncertainty affecting classification but, as we analyzed, they may not converge due to the myopic nature. 2) Our weighted-MOCU based method approximates the ELR method and still directly reduces the uncertainty affecting classification. Therefore it is more efficient, by prioritizing queries directly improving classification performance, than other converging methods that reduce total uncertainty, including BALD. This also has been shown empirically in different setups in our experiments.
> >
> > Q6. Complexity of class space.
> > A6. Consider a large class space with size $M$. In WMOCU function, the predictive distribution (line 22 in Algorithm 1) is calculated for $O(MN_xN_{\theta})$ times. In ACQUISITIONFUN, WMOCU is called for $M$ times. So the complexity of calculating the acquisition function is $O(M^2N_xN_{\theta})$.
> >
> > Q7. Performance on different noise level
> > A7. Data noise levels can affect the efficiency of different active learning methods. We have run preliminary tests on different noise levels. It is clear for some methods, such as MES, depending on uncertainty of candidates, the performance can degrade significantly with high noise. For the other methods, their performances are not that sensitive to the noise level and the performance trends are similar to the random sampling benchmark.
> >
> > Q8. Performance gap between ELR and weighted MOCU.
> > A8. Regarding the large gap between ELR and WMOCU in Fig. S8, obtained from real-world datasets, in contrast with the averaged performance with different true models in our synthetic experiments, it is expected that some methods perform much better than others, depending on the underlying feature-label distributions and data quality.
> >
> > We have explained the confusion term ‘side’ in the revision, and we have added error bars to our results to help better understand the performance trend.

---

### Author Response · Authors · 2020-11-23
**General response**

General response
We thank all the reviewers for their time and efforts in reviewing the paper and provide constructive suggestions. We have significantly revised our paper based on all four reviewers’ comments and we would like to highlight our major changes in our revision as follows:
1. We have reorganized the paper to improve the readability by rearranging the lemmas, proofs, and algorithms presentation with complexity analysis as suggested.
2. We have revised Section 3.2 to provide clearer analysis on the myopic issue of ELR methods.
3. We have added Fig. 2 and discussions in Sections 3.2 and 3.3 to intuitively illustrate the difference between MOCU and the weighted MOCU. The detailed setup for Fig. 2. is included in Appendix D.
4. In Section 3.3 we have generalized the weighting function as $1 - cK$ with an additional parameter $c$. This parameter can be used to balance the trade-off between short-term and long-term benefits of the proposed active learning method.
4. We have added a new experiment in Appendix G to compare ELR and weighted-MOCU based methods by showing the MOCU and acquisition function changes during the active learning procedures (Fig. S5). The experiment directly validates Lemma 4 and Theorem 1.
5. In Appendix G, we have also added another additional set of experiments to compare performance of different active learning methods under different noise levels (Fig. S6).

---

### Decision · Program_Chairs · 2021-01-07
**Final Decision**

**Decision:**

Accept (Poster)

**Comment:**

The paper proposed weighted-MOCU, a novel objective-oriented data acquisition criterion for active learning. The propositions are well-motivated, and all reviewers find the analysis of the drawbacks of several popular myopic strategies (e.g. ELR tends to stuck in local optima; BALD tends to be overly explorative)) interesting and insightful. Reviews also appreciate the novelty of the proposed weighted strategy for addressing the convergence issue of MOCU-based approaches. Overall I share the same opinions and believe the paper offers useful insights for the active learning community.

In the meantime, there were shared concerns among several reviewers in the readability (structure and intuition), lack of empirical results on more realistic active learning tasks, and limited discussion on the modeling assumptions. Although the rebuttal revision does improve upon many of these points, the authors are strongly encouraged to take into account the reviews, in particular, to further strengthen the empirical analysis and discussions, when preparing a revision.